# POINTLAM: LOCAL ATTENTIVE MAMBA FOR EFFICIENT POINT-BASED 3D OBJECT DETECTION

## ABSTRACT

3D object detection from LiDAR faces a fundamental trade-off between computational efficiency and the preservation of fine-grained geometries. The dominant voxel-based paradigm achieves efficiency by quantizing massive point clouds, but at the cost of inevitable information loss. Conversely, point-based methods excel at capturing precise geometries by directly processing raw points, yet have been constrained by the prohibitive complexity of their core operators for downsampling and spatial feature modeling. In this work, we tackle this dillema by introducing PointLAM, a novel framework for point-based 3D object detection that excels both in performance and efficiency. We systematically address the long-standing bottlenecks in point-based models through two synergistic designs. First, we propose a Dynamic Point Sampler (DPS) that intelligently curates an information-rich and structurally representative subset of raw points. It leverages a novel Deviation Network (DevNet) to capture each point's local distinctiveness, followed by a Doubly Sorted Sampling (DSS) strategy that retains the most informative points to reduce the workload of the 3D backbone. Second, our 3D backbone synergizes Bi-Directional Mamba (BDM) layers for global context modeling, and novel, lightweight Local Multiplicative Aggregation (LMA) layers for efficiently capturing intricate local geometries without computationally expensive neighborhood queries. Extensive experiments show that PointLAM sets a new benchmark for efficient point-based 3D object detection. On both nuScenes and Waymo datasets, PointLAM not only significantly surpasses prior point-based models but also achieves comparable performance against strong voxel-based competitors like LION and DSVT. Crucially, these competitive results are achieved with a fraction of the model parameters and latency, demonstrating a superior balance between accuracy and efficiency.

## 1 INTRODUCTION

Accurate 3D object detection from LiDAR point clouds, a cornerstone of modern autonomous systems, hinges on effectively processing vast and unstructured raw data. To this end, the field has largely diverged into two primary categories: voxel-based and point-based methods. Voxel-based approaches have become the de facto standard, regularizing irregular points into structured volumetric grids, which are then processed by 3D Sparse Convolutional Neural Networks (SpCNNs) (Fan et al., 2024; Zhang et al., 2024a; 2023; Chen et al., 2023b;a; Shi et al., 2023; Fan et al., 2022b; Yin et al., 2021; Shi et al., 2020b) or Transformers (Wu et al., 2024a; Liu et al., 2024c; Wang et al., 2023; Liu et al., 2023; Bai et al., 2022; Zhou et al., 2022; Mao et al., 2021). This efficiency, however, is achieved by sacrificing geometric fidelity. In contrast, point-based methods (Shi et al., 2019; Yang et al., 2020; Qi et al., 2018) operate directly on raw point sets, endowing them with the intrinsic ability to preserve precise,

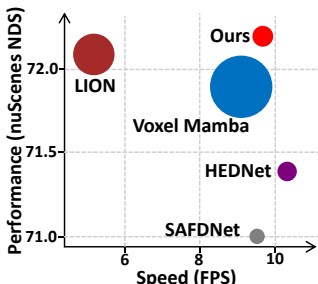

Figure 1: Comparison of the performance, speed, and parameter count (represented by circle size) of state-of-the-art 3D object detectors.

Figure 2: Illustration of the point cloud processing schemes adopted by point-set-based (Qi et al., 2017a;b; Shi et al., 2019; Yang et al., 2020) and voxel-based 3D object detectors (Liu et al., 2024b; Zhang et al., 2023; 2024a; Wang et al., 2023), as well as our proposed efficient point-based solution.

fine-grained geometric details, avoiding the quantization errors and information loss inherent to voxelization.

However, this theoretical advantage has been consistently overshadowed by systemic inefficiency, hindering the widespread adoption of point-based detectors. The challenge manifests across the entire pipeline, beginning with the critical bottleneck of downsampling the massive input point cloud. Computationally intensive methods like Farthest Point Sampling (FPS) (Yang et al., 2020; Liang et al., 2024; Han et al., 2024; Shi et al., 2020a; 2023), while effective, suffer from significant latency, while simple alternatives like Random Sampling risk discarding crucial structural information. Beyond sampling, classic point-based backbones struggle to capture both fine-grained local patterns and long-range contextual dependencies without expensive neighborhood queries (e.g., k-NN) or complex hierarchical structures (Qi et al., 2017a;b; Shi et al., 2019; Yang et al., 2020). This efficiency gap has cultivated a clear bias in the field, as even recent highly efficient architectures incorporating Mamba (Gu & Dao, 2023) have exclusively favored voxel-based frameworks (Zhang et al., 2024b; Liu et al., 2024b). To our best knowledge, relevant attempts on the point-based paradigm have been notably absent from recent literature. Given the precision benefits of direct point processing, developing an efficient and powerful point-based paradigm is not just important, but essential for the future of 3D perception.

This paper introduces PointLAM, a novel direct point-based 3D object detection architecture tailored to address these long-standing limitations in existing point-based solutions. To resolve the efficiency-precision dilemma of informative point cloud downsampling, we propose the Dynamic Point Sampler (DPS). Instead of relying on computationally prohibitive sampling or information-lossy quantization, DPS adopts a two-stage strategy. It initially partitions the point cloud into local regions and employs a Deviation Network (DevNet) to compute a feature for each point that captures its distinctiveness within its local neighborhood. Subsequently, a Doubly Sorted Sampling (DSS) mechanism uses these features to select a structurally representative subset of points efficiently. This process dynamically and efficiently preserves the most salient geometric and contextual points for the backbone.

For the challenging task of capturing fine-grained local geometries directly from sparse points at low computational costs, PointLAM's backbone synergistically combines local and global feature extraction. For the global context, we employ Bi-Directional Mamba (BDM) layers to effectively model long-range dependencies with linear complexity. For local feature interaction, we introduce the Local Multiplicative Aggregation (LMA) layer. To circumvent the high cost of neighborhood queries like k-NN, LMA leverages the discrete grid structure to implicitly define spatial neighborhoods. This novel approach enables a sparse convolutional operator to perform a weighted aggregation directly over the features of neighboring points, efficiently achieving local geometric interactions while ensuring each point maintains its individuality throughout the process. Furthermore, this is complemented by a lightweight element-wise multiplication branch to enhance the feature modeling capacity with minimal overhead.

By integrating these efficient designs, PointLAM achieves significant efficiency advantages over strong voxel-based methods such as LION (Liu et al., 2024b) and DSVT (Wang et al., 2023) with comparable 3D object detection performance on established benchmarks. Notably, PointLAM achieves competitive performance on both nuScenes and Waymo datasets while involving only a fraction of the parameters, operations, and latency. To summarize, the main contributions of this paper include:

- We introduce the Dynamic Point Sampler (DPS) to resolve the point cloud downsampling bottleneck. Leveraging a Deviation Network (DevNet) to gauge point distinctiveness and

a Doubly Sorted Sampling (DSS) mechanism for efficient selection, DPS curates a highly informative point set for the backbone.

- We propose the Local Multiplicative Aggregation (LMA) layer, a novel and efficient local aggregator. Designed to complement the global context modeling of Bi-Directional Mamba (BDM) layers, LMA captures intricate geometric patterns by leveraging sparse convolutions on a transient grid, thus circumventing expensive neighborhood queries.

- We present the streamlined PointLAM framework, a novel point-based 3D object detection architecture. It directly processes raw points through DPS and builds its 3D feature extraction backbone upon the proposed PointLAM blocks, which feature a synergy between advanced local geometry extraction and global contextual modeling.

- PointLAM achieves competitive performance on established 3D object detection benchmarks. As a point-based method, it is even more efficient than the strong and highly efficient voxel-based competitors LION and DSVT, with only a fraction of their complexity in terms of parameters, operations, and latency costs.

## 2 RELATED WORK

### 2.1 LiDAR-BASED 3D OBJECT DETECTION

LiDAR-based 3D object detectors can be broadly categorized into two types: point-based and voxel-based methods. Point-based methods (Qi et al., 2017a;b; Shi et al., 2019; Yang et al., 2020; Qi et al., 2018) directly process raw, irregular point sets using techniques such as the PointNet series (Qi et al., 2017a;b) to extract geometric features from local point neighborhoods. While preserving precise location information, these methods often contend with challenges such as high computational costs for sampling and grouping, lower inference efficiency, and difficulties in capturing expansive contextual features due to their localized processing. In contrast, voxel-based approaches (Wang et al., 2023; Fan et al., 2022a; Bai et al., 2022; Chen et al., 2023b; Zhang et al., 2023) convert unstructured point clouds into regular 3D voxel grids. This regularization allows the application of more conventional network architectures and has led to voxel-based methods becoming a mainstream approach in 3D object detection.

Voxel-based methods can be broadly categorized into Sparse Convolutional Neural Network (SpCNN)-based and Transformer-based. SpCNN-based methods (Yin et al., 2021; Chen et al., 2023b; Li et al., 2023b; Fan et al., 2024; Zhang et al., 2024a; 2023; Chen et al., 2023a; Shi et al., 2020b; Yan et al., 2018; Deng et al., 2021; Fan et al., 2022b) utilize 3D sparse convolutions, which efficiently process only non-empty voxels. While efficient for sparse data, the reliance on small kernels can restrict the effective receptive field, limiting the capture of long-range dependencies. Transformer-based methods (Bai et al., 2022; Fan et al., 2022a; Wang et al., 2023; Liu et al., 2024c; Dong et al., 2022; Li et al., 2023a; Sun et al., 2022; Zhou et al., 2022; Liu et al., 2023; Wu et al., 2024a; Zhu et al., 2023; Mao et al., 2021; He et al., 2022) have been introduced to the voxel domain, grouping voxels and applying self-attention to model global relationships. However, the quadratic complexity of Transformers often necessitates processing a limited number of voxels or voxel groups to remain computationally feasible. The constraints of restricted receptive fields in SpCNNs and the computational demands or limited scope of voxel-based Transformers motivate the exploration of alternative architectures like Mamba, which PointLAM leverages within its point-based framework to capture global context efficiently.

### 2.2 MAMBA FOR 3D POINT CLOUD PROCESSING

Mamba (Gu & Dao, 2023) has recently been introduced into deep neural networks as a compelling alternative to Transformers. Mamba incorporates input-dependent parameters and a selection mechanism to achieve linear-time sequence modeling and strong performance. Its success has spurred adaptations to general vision tasks such as image classification, semantic segmentation, and 2D object detection, with models such as Vision Mamba (Zhu et al., 2024) and Vmamba (Liu et al., 2024a) employing different bidirectional SSMs or 2D-selective scanning techniques to process image data and learn global visual clues effectively.

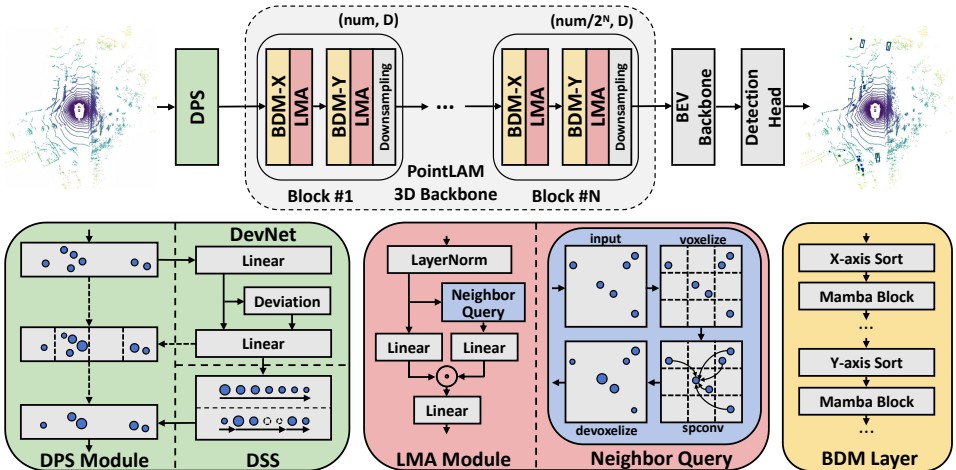

Figure 3: The overall architecture of PointLAM. **Top:** The pipeline starts with the Dynamic Point Sampler (DPS) downsampling raw points, followed by a 3D backbone of $N$ blocks. Each block cascades X-axis or Y-axis Bi-Directional Mamba (BDM) layers, each paired with a Local Multiplicative Aggregation (LMA) module, followed by BEV projection and detection heads. **Bottom-Left:** DPS leverages a Deviation Network (DevNet) to encode distinctiveness and Doubly Sorted Sampling (DSS) to retain informative points. **Bottom-Center:** LMA captures local geometry by utilizing a transient voxel grid for neighbor indexing and kernel-based weighted aggregation, refined by adaptive multiplicative modulation. **Bottom-Right:** BDM models global context by sequentially processing points sorted along the X and Y axes via Mamba blocks.

Building on these advancements, leveraging Mamba for 3D point cloud analysis is an emerging research frontier, initiated by pioneering efforts such as PointMamba (Liang et al., 2024). Applying these inherently sequential models to unordered and unstructured 3D point clouds presents distinct challenges: devising effective point serialization strategies that preserve spatial locality, adequately capturing fine-grained local geometric details (which is not the primary strength of global-centric Mamba), and addressing the non-causal nature of 3D spatial relationships. Subsequent works like Mamba3D (Han et al., 2024), Point cloud mamba (Zhang et al., 2025), and others (*e.g.*, Liu et al. (2024b); Zhang et al. (2024b)) have explored various solutions, often involving specific serialization protocols, such as coordinate sorting (Liu et al., 2024a) and space-filling curves (Sagan, 1993; Orenstein, 1986), specialized scanning methods, or the integration of components such as convolutions (Liu et al., 2024b; Zhang et al., 2024b) to boost local feature extraction.

## 3 METHOD

This section details the architecture of PointLAM, our proposed efficient point-based 3D detector. We begin with an overview of the overall framework (Sec. 3.1), followed by in-depth descriptions of its two core innovations: the Dynamic Point Sampler (Sec. 3.2) and the PointLAM backbone block (Sec. 3.3).

### 3.1 OVERALL ARCHITECTURE

As illustrated in Fig. 3, PointLAM processes raw point clouds through a streamlined pipeline comprising the Dynamic Point Sampler (DPS) and a hierarchical 3D backbone. First, the proposed Dynamic Point Sampler (DPS) (Sec. 3.2) efficiently downsamples the massive input. It utilizes a Deviation Network (DevNet) to encode geometric distinctiveness for each point, followed by a Doubly Sorted Sampling (DSS) strategy to retain the most informative points based on these deviation features. The sampled points are then processed by the 3D backbone, which consists of a stack of $N$ PointLAM blocks. To synergize global and local modeling, each block sequentially integrates Bi-Directional Mamba (BDM) layers (Sec. 3.3.2) for long-range dependency modeling and Local Multiplicative Aggregation (LMA) modules (Sec. 3.3.1) for fine-grained feature extraction. Notably, LMA circumvents expensive neighbor queries by utilizing a transient voxel grid to define

spatial neighborhoods for efficient kernel-based feature aggregation. Finally, after stage-wise random downsampling, the point features are projected into a Bird's-Eye-View (BEV) representation for the detection head.

## 3.2 Dynamic Point Sampler

Point-based 3D detectors typically begin by downsampling the input cloud to a manageable size. While methods like Farthest Point Sampling (FPS) (Yang et al., 2020; Liang et al., 2024; Han et al., 2024; Shi et al., 2020a; 2023) are popular for ensuring uniform spatial coverage, their iterative nature imposes a significant computational bottleneck, hindering real-time applications. To overcome this, we introduce the Dynamic Point Sampler (DPS), a highly efficient two-stage pipeline designed to selectively preserve the most informative points. The DPS consists of a Deviation Network (DevNet) for discriminative feature encoding, followed by a Doubly Sorted Sampling (DSS) algorithm for rapid, saliency-based point selection.

**Deviation Network.** The goal of DevNet is to produce augmented point features that explicitly encode each point's distinctiveness within its local context, thereby providing a strong basis for subsequent sampling. The process begins by partitioning the input point cloud into a grid of non-overlapping local regions. For each point $p_i$ within a region, we form an initial feature vector $F_{p_i}$ by concatenating its raw coordinates ($F_{\text{coords}}$), its offset from the region's centroid ($F_{\text{cluster}}$), and its offset from the region's geometric center ($F_{\text{center}}$).

These initial features $\{F_{p_i}\}_{i=1}^N$ for all $N$ points in a region are then passed through a shared linear layer to yield intermediate features $\{h_i\}_{i=1}^N$. Here lies the key difference from standard Point Feature Networks (PFNs) (Zhou & Tuzel, 2018), which typically use max-pooling for contextual aggregation. Instead, DevNet introduces a Deviation operation to explicitly encode the uniqueness of each point. This is achieved by calculating the deviation of each point's feature $h_i$ from the mean feature of its local region, $\bar{h} = \frac{1}{N}\sum_{i=1}^N h_i$. The final augmented feature $F'_{p_i}$ for each point is then created by concatenating the point's individual feature with its calculated Deviation:

$$F'_{p_i} = \text{Concat}(h_i, \underbrace{h_i - \bar{h}}_{\text{Deviation}}) \tag{1}$$

This Deviation-based feature design inherently highlights points that are unique or geometrically significant within their neighborhood, making them prime candidates for selection.

**Doubly Sorted Sampling.** With the rich features $F'_{p_i}$ generated by DevNet, the DSS algorithm performs the dual task of saliency scoring and efficient selection. First, for each point $p_i$, an importance score $S(p_i)$ is derived from its augmented feature vector, typically by taking the L2-norm of the channel values: $S(p_i) = ||F'_{p_i}||_2$.

Next, all points in the cloud are sorted in a two-step process: they are first sorted globally by their importance score $S(p_i)$ in descending order, and then stably sorted by their region indices. This ensures that higher-scoring points are always listed first. Let $p'_j$ denote the point at the $j$-th position in this doubly sorted list, with a region index of $r'_j$. The top-$k$ selection rule is then formulated as:

$$\text{Select } p'_j \iff \begin{cases} \text{True} & \text{if } j < k \\ r'_j \neq r'_{j-k} & \text{if } j \geq k \end{cases} \tag{2}$$

Essentially, this point selection rule picks the first $k$ points ($j < k$) from the doubly sorted list. For $j \geq k$, $p'_j$ is selected if its region index $r'_j$ differs from $r'_{j-k}$ (the region index of the point $k$ positions prior). This mechanism retains the $k$ highest-scoring points per region. As a result, the proposed DSS strategy efficiently preserves the $k$ most informative points per region, ensuring effective point cloud sparsification while preserving essential local structures for subsequent efficient feature extraction by our attentive PointLAM blocks.

## 3.3 PointLAM Block

The PointLAM block constitutes the core of PointLAM's feature extraction pipeline. It innovatively combines local feature extraction and global sequence modeling to achieve efficient and fine-grained processing of point cloud data. This section describes the two core sub-modules within a PointLAM block, namely Local Multiplicative Aggregation (LMA) and Bi-Directional Mamba (BDM).

### 3.3.1 LOCAL MULTIPLICATIVE AGGREGATION (LMA)

Capturing fine-grained local geometry is vital for 3D detection, yet traditional point-based methods (Qi et al., 2017a;b; Shi et al., 2019; Yang et al., 2020) that rely on explicit neighborhood queries (e.g., k-NN) often introduce significant computational overhead. To address this, we introduce the Local Multiplicative Aggregation (LMA) layer, a novel module that efficiently extracts rich local features.

**Neighbor Query.** To allow any point within the point cloud to efficiently interact with its spatial neighbors, we define its neighborhood implicitly via a discrete voxel grid. Formally, for each point $p_i$ with coordinates $\mathbf{p}_i$ and features $\mathbf{f}_i \in \mathbb{R}^D$, we first associate it with a unique integer voxel index $\mathbf{v}_i = \mathcal{V}(\mathbf{p}_i)$, where $\mathcal{V}$ is the quantization function. The neighborhood for point $p_i$, denoted $\mathcal{N}(i)$, is then defined as the set of all points $\{p_j\}$ whose voxel indices lie within a fixed-size kernel window (e.g., $3 \times 3 \times 3$) centered on $\mathbf{v}_i$. Local aggregation is then performed as a weighted summation over the features of these neighboring points. The weights are supplied by a learnable 3D convolutional kernel $W$, indexed by the relative offset between the points' voxel indices. This operation, which effectively simulates a sparse convolution directly on the point set, is formulated as:

$$\mathbf{f}_i^{\text{local}} = \sum_{j \in \mathcal{N}(i)} W(\mathcal{V}(\mathbf{p}_j) - \mathcal{V}(\mathbf{p}_i)) \cdot \mathbf{f}_j \qquad (3)$$

Crucially, as shown in Eq. 3, each point $p_i$ maintains its individuality. Point features are never merged or averaged into a voxel representation; the grid serves purely as a structural scaffold to facilitate efficient, convolution-based feature interaction between discrete points. The resulting context-rich features $\{\mathbf{f}_i^{\text{local}}\}$ are then passed through a linear layer to produce a local context representation $\mathbf{c}_i \in \mathbb{R}^D$.

**Adaptive Feature Modulation.** To adaptively refine the aggregated local features, we introduce a parallel modulation mechanism. The original input feature $\mathbf{f}_i$ is passed through a separate linear layer to produce a gating vector $\mathbf{g}_i \in \mathbb{R}^D$. For fusing this gating vector with the local content vector $\mathbf{c}_i$, we explored several aggregation modes. Here, we adopt element-wise multiplicative aggregation ($\odot$) for its superior expressive power. This interaction allows the gating branch ($\mathbf{g}_i$) to act as a dynamic, feature-wise filter, adaptively modulating the aggregated local features on a channel-by-channel basis. A residual connection is added to the modulated feature to stabilize training. The complete LMA operation is thus formulated as:

$$\mathbf{f}_i^{\text{out}} = (\mathbf{g}_i \odot \mathbf{c}_i) + \mathbf{f}_i \qquad (4)$$

This design is highly efficient. The neighbor query captures rich local context without any explicit neighborhood search, while the adaptive modulation stage provides a powerful yet lightweight mechanism to refine these features. This allows LMA to produce enhanced local representations with minimal computational overhead.

### 3.3.2 BI-DIRECTIONAL MAMBA

The Bi-Directional Mamba (BDM) layers in PointLAM capture long-distance dependencies in point cloud data with $O(N)$ computational complexity. Exploiting Mamba's selective aggregation capabilities, BDM allows relationships among distant points to be established, which complements the local context aggregation of the preceding LMA layer.

As shown in Fig.3, we order point clouds by primarily serializing points along the x-axis or y-axis, and critically, merging all samples within a batch into a single continuous sequence. The core implementation processes these ordered sequences along two primary spatial dimensions. The batch-merged, x-sorted points are processed through a Mamba block to capture dependencies along the x-direction. The resulting features are then reordered based on y-coordinates and processed through a second Mamba block to model y-direction relationships.

This bidirectional approach, built upon an efficient serialization strategy, allows BDM to effectively perceive 3D spatial structure from multiple dimensions without requiring computationally expensive global self-attention. By maintaining $O(N)$ complexity while modeling long-distance interactions, BDM enables efficient processing of large-scale point clouds. Note that while other serialization strategies are available, we choose axis scan for its simplicity. With local geometries well-captured by LMA layers, even a simple axis scan yields competitive performance, as will be demonstrated through ablation experiments.

Table 1: Performance comparison on the **validation** set of nuScenes dataset. 'C.V.', 'Ped.', 'M.C.', and 'T.C.' denote construction vehicle, pedestrian, motorcycle, and traffic cone, respectively.

| Method | Representation | NDS | mAP | Car | Truck | Bus | Trailer | C.V. | Ped. | M.C. | Bike | T.C. | Barrier |
|---|---|---|---|---|---|---|---|---|---|---|---|---|---|
| PillarNeXt (Li et al., 2023b) | Pillar | 68.4 | 62.2 | 85.0 | 57.4 | 67.6 | 35.6 | 20.6 | 86.8 | 68.6 | 53.1 | 77.3 | 69.7 |
| DSVT (Wang et al., 2023) | | 71.1 | 66.4 | 87.4 | 62.6 | 75.9 | 42.1 | 25.3 | 88.2 | 74.8 | 58.7 | 77.8 | 70.9 |
| CenterPoint (Yin et al., 2021) | Voxel | 66.5 | 59.2 | 84.9 | 57.4 | 70.7 | 38.1 | 16.9 | 85.1 | 59.0 | 42.0 | 69.8 | 68.3 |
| TransFusion-L (Bai et al., 2022) | | 70.1 | 65.5 | 86.9 | 60.8 | 73.1 | 43.4 | 25.2 | 87.5 | 72.9 | 57.3 | 77.2 | 70.3 |
| VoxelNeXt (Chen et al., 2023b) | | 66.7 | 60.5 | 83.9 | 55.5 | 70.5 | 38.1 | 21.1 | 84.6 | 62.8 | 50.0 | 69.4 | 69.4 |
| HEDNet (Zhang et al., 2023) | | 71.4 | 66.7 | 87.7 | 60.6 | 77.8 | 50.7 | 28.9 | 87.1 | 74.3 | 56.8 | 76.3 | 66.9 |
| FSDv2 (Fan et al., 2024) | | 70.4 | 64.7 | 83.7 | 51.6 | 66.4 | 59.1 | 32.5 | 87.1 | 71.4 | 51.7 | 80.3 | 78.7 |
| SAFDNet (Zhang et al., 2024a) | | 71.0 | 66.3 | 87.6 | 60.8 | 78.0 | 43.5 | 26.6 | 87.8 | 75.5 | 58.0 | 75.0 | 69.7 |
| LION (Liu et al., 2024b) | | 72.1 | **68.0** | 87.9 | 64.9 | 77.6 | 44.4 | 28.5 | 89.6 | 75.6 | 59.4 | 80.8 | 71.6 |
| Voxel Mamba (Zhang et al., 2024b) | | 71.9 | 67.5 | 87.9 | 62.8 | 76.8 | 45.9 | 24.9 | 89.3 | 77.1 | 58.6 | 80.1 | 71.5 |
| **PointLAM** (ours) | Point | **72.2** | 67.8 | 88.6 | 64.1 | 78.9 | 45.3 | 26.5 | 89.2 | 73.9 | 59.5 | 80.2 | 72.1 |

Table 2: Performance comparison on the **validation** set of Waymo Open Dataset (single-frame setting). Symbol '-' denotes that the result is not available.

| Method | Representation | ALL (3D mAPH) L1 | L2 | Vehicle (AP/APH) L1 | L2 | Pedestrian (AP/APH) L1 | L2 | Cyclist (AP/APH) L1 | L2 |
|---|---|---|---|---|---|---|---|---|---|
| PV-RCNN (Shi et al., 2020a) | Point-Voxel | 69.6 | 63.3 | 77.5 / 76.9 | 69.0 / 68.4 | 75.0 / 65.7 | 66.0 / 57.6 | 67.8 / 66.4 | 65.4 / 64.0 |
| PV-RCNN++ (Shi et al., 2023) | | 75.2 | 68.6 | 79.1 / 78.6 | 70.3 / 69.9 | 80.6 / 74.6 | 71.9 / 66.3 | 73.5 / 72.4 | 70.7 / 69.6 |
| PointPillar (Lang et al., 2019) | Pillar | 63.3 | 57.5 | 71.6 / 71.0 | 63.1 / 62.5 | 70.6 / 56.7 | 62.9 / 50.2 | 64.4 / 62.3 | 61.9 / 59.9 |
| PillarNet (Shi et al., 2022) | | 74.6 | 68.4 | 79.1 / 78.6 | 70.9 / 70.5 | 80.6 / 74.0 | 72.3 / 66.2 | 72.3 / 71.2 | 69.7 / 68.7 |
| SWFormer (Sun et al., 2022) | | - | - | 77.8 / 77.3 | 69.2 / 68.8 | 80.9 / 72.7 | 72.5 / 64.9 | - | - |
| PillarNeXt (Li et al., 2023b) | | 75.7 | 69.7 | 78.4 / 77.9 | 70.3 / 69.8 | 82.5 / 77.1 | 74.9 / 69.8 | 73.2 / 72.2 | 70.6 / 69.6 |
| FlatFormer (Liu et al., 2023) | | - | 67.2 | - | 69.0 / 68.6 | - | 71.5 / 65.3 | - | 68.6 / 67.5 |
| PTv3 (Wu et al., 2024a) | | - | 70.5 | - | 71.2 / 70.8 | - | 76.3 / 70.4 | - | 71.5 / 70.4 |
| SECOND (Yan et al., 2018) | Voxel | 63.1 | 57.2 | 72.3 / 71.7 | 63.9 / 63.3 | 68.7 / 58.2 | 60.7 / 51.3 | 60.6 / 59.3 | 58.3 / 57.1 |
| FSDv1 (Fan et al., 2022b) | | 77.3 | 70.8 | 79.2 / 78.8 | 70.5 / 70.1 | 82.6 / 77.3 | 73.9 / 69.1 | 77.1 / 76.0 | 74.4 / 73.3 |
| Part-A2 (Shi et al., 2020b) | | 70.3 | 63.8 | 77.1 / 76.5 | 68.5 / 68.0 | 75.2 / 66.9 | 66.2 / 58.6 | 68.6 / 67.4 | 66.1 / 64.9 |
| Centerpoint (Yin et al., 2021) | | - | 67.6 | 76.6 / 76.0 | 68.9 / 68.4 | 79.0 / 73.4 | 71.0 / 65.8 | 72.1 / 71.0 | 69.5 / 68.5 |
| SST (Fan et al., 2022a) | | - | - | 76.2 / 75.8 | 68.0 / 67.6 | 81.4 / 74.1 | 72.8 / 65.9 | - | - |
| CenterFormer (Zhou et al., 2022) | | 73.2 | 69.1 | 75.0 / 74.4 | 69.9 / 69.4 | 78.0 / 72.4 | 73.1 / 67.7 | 73.8 / 72.7 | 71.3 / 70.2 |
| VoxelNeXt (Chen et al., 2023b) | | 76.3 | 70.1 | 78.2 / 77.7 | 69.9 / 69.4 | 81.5 / 76.3 | 73.5 / 68.6 | 76.1 / 74.9 | 73.3 / 72.2 |
| DSVT (Wang et al., 2023) | | 78.2 | 72.1 | 79.7 / 79.3 | 71.4 / 71.0 | 83.7 / 78.9 | 76.1 / 71.5 | 77.5 / 76.5 | 74.6 / 73.7 |
| HEDNet (Zhang et al., 2023) | | 79.4 | 73.4 | 81.1 / 80.6 | 73.2 / 72.7 | 84.4 / 80.0 | 76.8 / 72.6 | 78.7 / 77.7 | 75.8 / 74.9 |
| SAFDNet (Zhang et al., 2024a) | | 79.2 | 73.2 | 80.2 / 79.7 | 72.2 / 71.8 | 79.9 / 76.9 | 76.8 / 72.6 | 79.1 / 78.1 | 76.2 / 75.2 |
| LION (Liu et al., 2024b) | | 80.1 | 74.0 | 80.3 / 79.9 | 72.0 / 71.6 | 85.8 / 81.4 | 78.5 / 74.3 | 80.1 / 79.0 | 77.2 / 76.2 |
| Voxel Mamba (Zhang et al., 2024b) | | 79.6 | 73.6 | 80.8 / 80.3 | 72.6 / 72.2 | 85.0 / 80.8 | 77.7 / 73.6 | 78.6 / 77.6 | 75.7 / 74.8 |
| **PointLAM** (ours) | Point | 79.7 | 73.6 | 80.0 / 79.6 | 71.7 / 71.3 | 85.1 / 81.1 | 78.0 / 74.1 | 79.4 / 78.3 | 76.5 / 75.5 |

## 4 EXPERIMENT

### 4.1 IMPLEMENTATION DETAILS

Our framework is implemented in the OpenPCDet codebase (Team, 2020). For the DPS, the local region size is set to $(0.3m, 0.3m, 0.25m)$ for the nuScenes dataset and $(0.32m, 0.32m, 0.1875m)$ for Waymo. In the DSS stage, we retain the top-$k$ most salient points per region, where we set $k = 2$. The main 3D backbone consists of a stack of $N = 4$ PointLAM blocks. Within each LMA module, the sparse convolution employs a kernel size of $3 \times 3 \times 3$. We train PointLAM for 36 epochs on nuScenes and 24 epochs on the Waymo dataset. Other hyperparameters, such as data augmentation and post-processing, follow the settings from DSVT (Wang et al., 2023). All experiments are conducted on NVIDIA A800 (80GB) GPUs. For more architectural and training details, please refer to our supplementary material.

### 4.2 MAIN RESULTS

**nuScenes.** As shown in Table 1, PointLAM match up or even outperform strong competitors by using only a fraction of their computational cost. Specifically, PointLAM achieves an NDS score of 72.2, outperforming all previous methods, including the recent strong Mamba-based models LION (+0.1 NDS) and Voxel Mamba (+0.3 NDS), as well as leading Transformer-based methods like DSVT (+1.1 NDS). In terms of mAP, PointLAM surpasses all other methods including Voxel Mamba and SAFDNet, and only falls short of LION by 0.2, yet being 2x faster and 2x lighter. These results show that PointLAM strikes the best performance-efficiency trade-off among all methods, highlighting its potential as a strong yet efficient point-based 3D object detection baseline.

**Waymo.** As shown in Table 2, PointLAM demonstrates highly competitive performance on the challenging Waymo Open Dataset. Specifically, it achieves 73.6 L2 mAPH, on par with the strong Mamba-based baseline of Voxel Mamba (73.6), and comparable to the state-of-the-art LION (74.0).

Table 3: Comparison of parameter count, computation cost, and inference latency of different 3D detection backbones.

| Method | Venue | Backbone | Representation | #Parameters (M) | FLOPs (G) | Latency (ms) | NDS | L2 mAPH |
|---|---|---|---|---|---|---|---|---|
| PVRCNN (Shi et al., 2020a) | CVPR'20 | | Point-Voxel | 8.5 | 47.4 | 281.2 | - | 63.3 |
| CenterPoint (Yin et al., 2021) | CVPR'21 | | Voxel | 2.7 | 41.8 | 60.1 | 66.5 | 67.6 |
| VoxelRCNN (Deng et al., 2021) | AAAI'21 | | Voxel | 11.7 | 23.3 | 94.5 | - | 66.2 |
| VoxelNeXt (Chen et al., 2023b) | CVPR'23 | spCNN | Voxel | 15.5 | 97.5 | 192.9 | 66.7 | 70.1 |
| PVRCNN++ (Shi et al., 2023) | IJCV'23 | | Point-Voxel | 9.5 | 55.4 | 97.3 | - | 68.6 |
| HEDNet (Zhang et al., 2023) | NeurIPS'23 | | Voxel | 4.6 | 106.2 | 96.7 | 71.4 | 73.4 |
| SAFDNet (Zhang et al., 2024a) | CVPR'24 | | Voxel | 3.7 | 60.8 | 104.3 | 71.0 | 73.2 |
| FlatFormer (Liu et al., 2023) | CVPR'23 | | Pillar | 1.1 | 48.1 | 62.3 | - | 67.2 |
| DSVT-Pillar (Wang et al., 2023) | CVPR'23 | Transformer | Pillar | 1.2 | 35.6 | 73.2 | 71.1 | 71.0 |
| DSVT-voxel (Wang et al., 2023) | CVPR'23 | | Voxel | 2.7 | 108.6 | 115.6 | - | 72.1 |
| Voxel Mamba (Zhang et al., 2024b) | NeurIPS'24 | | Voxel | 15.1 | 246.2 | 109.8 | 71.9 | 73.6 |
| LION (Liu et al., 2024b) | NeurIPS'24 | Mamba | Voxel | 10.1 | 165.8 | 195.3 | 72.1 | 74.0 |
| **PointLAM** (ours) | - | | Point | 5.0 | 90.7 | 104.9 | 72.2 | 73.6 |

It also surpasses a range of established 3D object detectors, including the spCNN-based HEDNet (73.4) and Transformer-based DSVT (72.1). Thanks to its point-based design that better preserves fine-grained geometries, PointLAM shows particular strength in detecting smaller objects such as "Pedestrian" and "Cyclist", where it outperforms Voxel Mamba. Perhaps more significant is that PointLAM achieves this superior performance as a point-based framework and with dramatically reduced parameter count, complexity, and latency, as discussed in the next section.

**Model Efficiency.** We compare PointLAM against other leading detectors in Table 3. We report the number of parameters, FLOPs (G), and inference latency (ms), alongside key performance metrics. All latency measurements are benchmarked on a single NVIDIA A800 GPU with a batch size of 1. Compared to the state-of-the-art method of LION, PointLAM achieves a higher NDS score while using approximately 50% fewer parameters, 45% fewer FLOPs, and running nearly twice as fast. Compared to Voxel Mamba, PointLAM delivers comparable L2 mAPH with 3x fewer parameters and 2.7x fewer FLOPs. Perhaps more significant is that PointLAM achieves such superior efficiency as a point-based detector, known to be inherently less efficient compared to voxel-based or pillar-based paradigms. PointLAM demonstrates for the first time that point-based methods can achieve both superior accuracy and competitive inference speeds for large-scale 3D object detection.

### 4.3 ABLATION STUDIES

To validate the effectiveness of our key design choices, we conduct a series of ablation studies. By default, experiments are performed on the nuScenes dataset with an 18-epoch schedule and on the 1/5 subsets of the Waymo Open Dataset with a normal schedule.

**Effectiveness of Dynamic Point Sampler.** We validate the two core stages of our Dynamic Point Sampler (DPS) in Table 4: the Encoding stage for point-level feature learning, and the Sampling stage for distilling these features. Our baseline, representative of standard voxelization, uses a Point Feature Network (PFN) for encoding with Pooling for sampling, and achieves 71.42 NDS. From this baseline, replacing the PFN with our proposed DevNet for encoding boosts performance to 71.59 NDS, confirming the effectiveness of our deviation-based features. Separately, replacing Pooling with our DSS sampling method also improves performance to 71.56 NDS, validating its superiority over simple aggregation. Finally, when combining our DevNet for encoding with DSS for sampling, the model yields the best performance (71.82 NDS), highlighting the powerful synergy between learning discriminative features and intelligently selecting them.

**Effectiveness of backbone components.** Table 5 ablates the effect of key designs within the Point-LAM block. First, we observe that using bi-directional BDM (scanning along both X and Y axes) yields a clear improvement of +0.9 NDS over uni-directional BDM. The results reveal a marked synergy between our local (LMA) and global (BDM) modelling modules: While both demonstrate strong performance individually, combining them leads to further performance gains. Specifically,

Table 4: Ablation on designs in Dynamic Point Sampler.

| Encoding | Sampling | mAP | NDS |
|---|---|---|---|
| PFN | Pooling | 67.41 | 71.42 |
| DevNet | Pooling | 67.82 | 71.59 |
| PFN | DSS | 67.77 | 71.56 |
| DevNet | DSS | **68.14** | **71.82** |

Table 5: Ablation on core backbone components.

| BDM X-axis | BDM Y-axis | LMA | mAP | NDS |
|---|---|---|---|---|
| ✓ | | | 63.67 | 68.68 |
| | | ✓ | 64.37 | 69.40 |
| ✓ | | ✓ | 67.26 | 71.23 |
| ✓ | ✓ | | 64.87 | 69.58 |
| ✓ | ✓ | ✓ | **68.14** | **71.82** |

Table 6: Effect of different aggregation modes in LMA.

| Aggregation Mode | mAP | NDS |
|---|---|---|
| Addition | 67.13 | 70.98 |
| Subtraction | 67.51 | 71.53 |
| Multiplication | **68.14** | **71.82** |
| Concatenation | 67.63 | 71.51 |

adding the LMA module to the bi-directional BDM boosts performance by a remarkable +2.24 NDS. This significant gain confirms that our efficient LMA module effectively captures fine-grained local patterns that complement the long-range dependencies modeled by the Mamba backbone.

**Effectiveness of Aggregation Mode in LMA.** We analyze our choice of aggregation mode within the LMA module in Table 6. The results show that our chosen Multiplication mode significantly outperforms other aggregation strategies. It achieves 71.82 NDS, surpassing the next-best alternative by a notable margin of +0.29 NDS. This result validates our design choice. Our parallel modulation branch, using the computationally negligible element-wise multiplication operator, forms a dynamic feature-wise filter that adaptively recalibrates local features with minimal overhead of a single linear layer. This synergy between the targeted design and efficient operator enables a significant performance gain, establishing our approach as a highly efficient and effective solution for local modeling.

**Effectiveness of Importance-Based Sampling** We conduct ablations on 1/10 Waymo subset to understand the validity of our feature-based importance scoring. In Table 7, we compare our default method using feature-based importance-score sorting (dubbed "DSS-IS") against a variant that uses random shuffle ("DSS-Random") across varying $k$ values (the number of points sampled per region). We notice that: First, our importance-based "DSS-IS" consistently outperforms the random selection variant by considerable margins of +0.4 to +0.6 L1 mAPH, corroborating its effectiveness. Second, the proposed sampling technique is reasonably robust to $k$, making it an easily generalizable plug-and-play module for any point-based methods.

Table 7: Ablation on importance-based scoring and sensitivity to $k$.

| Method | L1 mAPH | L2 mAPH |
|---|---|---|
| DSS-Random (k=1.0) | 74.59 | 68.50 |
| **DSS-IS (k=1.0)** | **75.02** | **68.90** |
| DSS-Random (k=1.5) | 74.57 | 68.48 |
| **DSS-IS (k=1.5)** | **74.99** | **68.91** |
| DSS-Random (k=2.0) | 74.59 | 68.46 |
| **DSS-IS (k=2.0)** | **75.14** | **69.02** |
| DSS-Random (k=2.5) | 74.51 | 68.39 |
| **DSS-IS (k=2.5)** | **74.93** | **68.79** |

**Effectiveness of different Mamba scan orders.** We compare Axis Sort, which simply sorts points along X then Y axes, against popular alternatives in Table 8. These alternatives include a single fixed random sequence ("Random"), complex geometry-aware space-filling curves ("Hilbert" / "Z-order"), and block-wise alternating / random selection of Hilbert / Z-order ("Shuffle" / "Shuffle*"). As expected, the costly "Hilbert" and "Z-order" scans perform marginally better by preserving spatial locality, but are over 50x slower than our choice

Table 8: Effectiveness of different Mamba scan orders.

| Scan order | mAP | NDS | Latency (ms) |
|---|---|---|---|
| Random | 66.74 | 71.08 | 0.1 |
| Hilbert | 67.06 | **71.26** | 5.5 |
| Z-order | **67.27** | 71.23 | 5.6 |
| Shuffle | 66.35 | 70.62 | 11.2 |
| Shuffle* | 66.75 | 70.89 | 11.2 |
| Axis Sort | 67.26 | 71.23 | 0.1 |

of simple Axis Short. Intriguingly, even the geometrically agnostic "Random" scan achieves highly competitive performance. PointLAM's robustness reinforces the validity of our synergistic backbone design, where LMA layers capture rich local features before serialization, rendering the BDM's global modeling task less sensitive to the precise point order. As such, we opt for the simple Axis Sort for a better precision-efficiency trade-off.

**Effectiveness of different Deviation mode of DevNet.** We investigated different operators for computing point distinctiveness within the Deviation Network. As shown in Table 9, the Subtraction mode yields the highest performance (75.63 L1 mAPH). This confirms that explicitly modeling the feature offset ($h_i - \bar{h}$) between a point and its local center effectively captures the geometric saliency required for informative sampling, outperforming simple addition or multiplication fusion.

**Effectiveness of Neighbor Query in LMA.** We compared our transient grid-based neighbor query against traditional KNN and the serialization-based query mechanism used in PTv3 Wu et al. (2024b), with all methods configured to aggregate the same number of neighbors per point. As shown in Table 10, our design achieves the best accuracy (75.63 L1 mAPH) with negligible latency (2.7 ms). In contrast, KNN is prohibitively slow (44.5 ms). While PTv3 is faster than KNN, its reliance on computing space-filling curves (e.g., Hilbert/Z-order) and reordering points introduces

Table 9: Ablation on deviation modes in DevNet.

| Deviation Mode | L1 mAPH | L2 mAPH |
|---|---|---|
| Subtraction | **75.63** | **69.54** |
| Addition | 75.54 | 69.45 |
| Multiplication | 75.49 | 69.39 |

Table 10: Ablation on neighbor queries in LMA.

| Neighbor | L1 mAPH | L2 mAPH | Latency (ms) |
|---|---|---|---|
| KNN | 74.61 | 68.40 | 44.5 |
| PTv3 | 72.52 | 66.53 | 11.2 |
| Ours | **75.63** | **69.54** | **2.7** |

Table 11: Ablation on kernel sizes in LMA.

| Kernel | L1 mAPH | L2 mAPH | Latency (ms) |
|---|---|---|---|
| $3 \times 3$ | 75.63 | 69.54 | 90.7 |
| $5 \times 5$ | 75.95 | 69.87 | 104.8 |
| $7 \times 7$ | 76.02 | 69.95 | 118.9 |

Table 12: Robustness comparison under varying input point densities.

| Density | Ours | | LION(Liu et al., 2024b) | |
|---|---|---|---|---|
| | L1 mAPH | L2 mAPH | L1 mAPH | L2 mAPH |
| 1/2 | 70.06 | 63.78 | 69.47 | 63.23 |
| 1/3 | 62.28 | 56.17 | 61.52 | 55.44 |
| 1/4 | 56.70 | 50.82 | 55.37 | 49.55 |
| 1/8 | 40.52 | 35.80 | 39.23 | 34.57 |
| 1/16 | 25.08 | 21.86 | 23.72 | 20.60 |

Table 13: Ablation on neighbor queries across different object sizes.

| Size ($m^3$) | Ours | | KNN | |
|---|---|---|---|---|
| | L1 mAPH | L2 mAPH | L1 mAPH | L2 mAPH |
| 1.0 | 3.75 | 3.28 | 2.51 | 2.17 |
| 1.5 | 20.70 | 18.82 | 19.09 | 17.27 |
| 2.0 | 29.75 | 27.27 | 27.77 | 25.34 |
| 2.5 | 35.40 | 32.69 | 33.76 | 31.08 |
| 3.0 | 47.12 | 43.97 | 43.72 | 40.68 |

overhead (11.2 ms). This validates that our indexing via a transient grid is a superior, lightweight solution for large-scale detection compared to both explicit search and complex serialization.

**Effectiveness of Kernel Size in LMA.** Table 11 explores the impact of kernel sizes in the LMA module. While increasing the kernel size from $3 \times 3$ to $7 \times 7$ brings marginal performance gains ($\sim$0.4 L1 mAPH), it significantly increases the model latency (from 90.7 ms to 118.9 ms). We adopt the $3 \times 3$ kernel as the default, striking the optimal balance between lightweight model design and strong detection performance.

**Robustness to Point Density.** We evaluated model robustness by testing on input point clouds with varying densities. To simulate varying densities, we employ interval sampling on the raw input sequences (ordered by laser scan lines), which ensures a globally uniform downsampling. As shown in Table 12, PointLAM consistently outperforms the voxel-based state-of-the-art LION Liu et al. (2024b) across all density levels. This confirms that our Dynamic Point Sampler (DPS) preserves critical geometry more effectively than voxelization, particularly in extreme sparsity scenarios where voxel-based methods suffer significant information loss.

**Performance on Different Object Sizes.** We compared our transient grid-based query (Ours) against a KNN baseline across varying object sizes (Table 13). Ours consistently excels, with the most substantial gains on small objects (Size 1.0). This stems from the aggregation logic: unlike KNN, which enforces a fixed neighbor count and risks feature contamination by including distant background noise for sparse targets, our approach strictly aggregates within a fixed spatial range. This excludes irrelevant noise, preserving pure feature for fine-grained detection.

**Interpretability of DPS and LMA.** We visualize the sampling points and feature activation in Figure 4. The contrast between retained points and discarded raw points highlights the selectivity of DPS: it aggressively filters out redundant, flat road surfaces while preserving the complete geometric skeleton of the foreground object. Furthermore, the feature heatmaps reveal that LMA learns a distinct geometry-aware representation. High activations align perfectly with structural boundaries, vehicle chassis, and corners, whereas feature responses on flat surfaces remain suppressed.

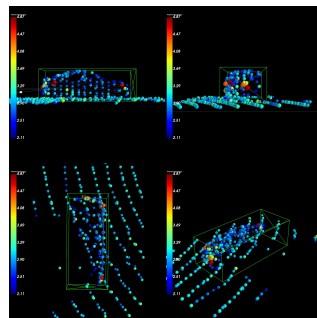

Figure 4: Visualization of DPS sampling and LMA feature learning. Raw input points are rendered in light gray, while points retained by DPS are colored based on the LMA feature magnitude.

## 5 CONCLUSION

In this work, we introduced PointLAM, a strong yet highly efficient point-based 3D object detection framework that addresses the long-standing dilemma between geometric precision and computational efficiency. It features a Dynamic Point Sampler (DPS), using a Deviation Network (DevNet) and Doubly Sorted Sampling (DSS) to intelligently select points, and a synergistic backbone where Local Multiplicative Aggregation (LMA) layers efficiently capture local patterns to complement the global modeling of Bi-Directional Mamba (BDM) layers. Extensive experiments on nuScenes and Waymo datasets validate our approach, demonstrating that PointLAM achieves competitive performance, including a top NDS score on nuScenes, while operating with a fraction of the computational cost of leading voxel-based competitors such as LION and DSVT. Overall, we believe that PointLAM represents a significant effort in improving the efficiency of point-based 3D object detectors. It is our hope that this work will inspire future endeavors in the point-based detection paradigm.

## 6 ETHICS STATEMENT

This research aims to advance 3D object detection to improve safety in applications like autonomous driving. We exclusively use public, anonymized academic benchmarks (nuScenes and Waymo datasets). We acknowledge that model performance is contingent on the distribution of the training dataset and may exhibit biases; therefore, comprehensive testing across diverse scenarios is required to ensure fairness and robustness before any real-world deployment. Given the safety-critical nature of this application, where perception failures can have severe consequences, this work is presented as foundational research that requires rigorous, domain-specific validation prior to any practical use.

## 7 REPRODUCIBILITY STATEMENT

Our implementation is built upon the publicly available OpenPCDet codebase and follows its standard data processing pipelines for the nuScenes and Waymo datasets. The architectural details and specific implementations of our novel modules—the Dynamic Point Sampler (DPS), Local Multiplicative Aggregation (LMA), and the Bi-Directional Mamba (BDM) Layer—are described in Sec. 3, which includes mathematical formulations for each key operation. All hyperparameters and training configurations are provided in Sec. 4.1, with further elaborations in the supplementary material. To ensure the full reproducibility of our work, we will release our complete source code upon publication.

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

# PointLAM: Local Attentive Mamba for Efficient Point-based 3D Object Detection —Supplementary Material

This material includes supplementary information on the preliminary background (Sec. A), more implementation details for the proposed framework (Sec. B), and further analytical experiments on the effectiveness and efficiency of our proposed model (Sec. D).

## A  BACKGROUND

### A.1  MAMBA

Mamba Gu & Dao (2023) is a type of selective State Space Model (SSM) designed for efficient sequence modeling. An SSM maps an input signal $x(t) \in \mathbb{R}^L$ to an output $y(t) \in \mathbb{R}^L$ via a latent state $h(t) \in \mathbb{R}^N$. The continuous-time formulation is:

$$h'(t) = \mathbf{A}h(t) + \mathbf{B}x(t)$$
$$y(t) = \mathbf{C}h(t) + \mathbf{D}x(t) \tag{5}$$

where $\mathbf{A} \in \mathbb{R}^{N \times N}$ denotes the state evolution matrix, $\mathbf{B} \in \mathbb{R}^{N \times 1}$, $\mathbf{C} \in \mathbb{R}^{1 \times N}$ are projection matrices, and $\mathbf{D} \in \mathbb{R}^1$ is a residual connection.

To handle discrete sequences (e.g., $x_k$), the continuous parameters $\mathbf{A}$ and $\mathbf{B}$ are discretized using a timescale parameter $\Delta$ and the Zero-Order Hold (ZOH) method: $\overline{\mathbf{A}} = \exp(\Delta\mathbf{A})$, $\overline{\mathbf{B}} = (\Delta\mathbf{A})^{-1}(\exp(\Delta\mathbf{A}) - \mathbf{I})\Delta\mathbf{B}$, where $\mathbf{I}$ is the identity matrix. The discretized SSM then follows the recurrence:

$$h_k = \overline{\mathbf{A}}h_{k-1} + \overline{\mathbf{B}}x_k$$
$$y_k = \overline{\mathbf{C}}h_k + \overline{\mathbf{D}}x_k \tag{6}$$

Mamba's key innovation is its selection mechanism (S6), which makes the parameters $\mathbf{B}$, $\mathbf{C}$, and $\Delta$ functions of the input sequence $x$. This input-dependency makes Mamba effectively time-varying and selective, enabling it to model complex dependencies with enhanced contextual awareness.

### A.2  DATASETS AND EVALUATION METRICS

**nuScenes.** nuScenes Caesar et al. (2020) is a large-scale outdoor 3D perception dataset containing 1,000 scenes captured at 2Hz within a 50m range (split into 750 for training, 150 for validation, and 100 for testing). It provides annotated bounding boxes for 10 traffic object categories of various sizes. The standard evaluation metrics for 3D object detection are mean Average Precision (mAP) and the nuScenes Detection Score (NDS). mAP accounts for classification accuracy and bounding box overlap, whereas NDS further considers multiple quality-related factors, including localization, orientation, attribute, and velocity errors.

**Waymo Open Dataset (WOD).** WOD Sun et al. (2020) is one of the largest outdoor 3D perception benchmarks. It comprises roughly 230,000 annotated frames (160k/40k/30k train/val/test split), each covering a $150m \times 150m$ area. For the evaluation of 3D object detection performance, it employs mean Average Precision (mAP) and its heading-weighted variant (mAPH) at two difficulty levels: Level 1 for objects with over five LiDAR points and Level 2 for objects with at least one point.

## B  FURTHER IMPLEMENTATION DETAILS

We describe further details on PointLAM's implementation and configurations. We also present the PyTorch-style pseudo-code for the implementation of the proposed DSS layer.

---

**Algorithm A** Simplified PyTorch-style pseudo-code for the masking operation of DSS

---

```
1: # sorted_region_indices: region indices first sorted by importance score and then stably sorted by region
   indices
2: # N: total number of points
3: # k: number of selected points per region
4: mask = torch.ones(size=N, dtype=torch.bool)
5: indices_j = sorted_region_indices[k:]
6: indices_j_minus_k = sorted_region_indices[:-k]
7: mask[k:]  = (indices_j != indices_j_minus_k)
8: return mask
```

---

## B.1 FRAMEWORK

Our PointLAM framework is implemented based on the OpenPCDet codebase (Team, 2020). The point feature dimension is 128 throughout the network, aligning with established voxel-based methods (Wang et al., 2023; Liu et al., 2024b; Zhang et al., 2024b). The following details pertain to configurations used for the Waymo (Sun et al., 2020) and nuScenes (Caesar et al., 2020) datasets. The input point cloud range is set to $[-75.52m, -75.52m, -2m, 75.52m, 75.52m, 4.0m]$ for Waymo and $[-54.0m, -54.0m, -5.0m, 54.0m, 54.0m, 3.0m]$ for nuScenes. Common preprocessing for both datasets involves masking points and bounding boxes that fall outside these predefined ranges and shuffling the points during both training and testing phases. Furthermore, a suite of common data augmentation techniques is applied to both datasets, including Ground-Truth (GT) Sampling, random world flips, random world rotations, random world scaling, and random world translation. For training, all experiments utilized a batch size of 4 per GPU. We use the AdamW optimizer with a one-cycle learning rate schedule, setting the peak learning rate to $3 \times 10^{-3}$. For nuScenes, training is performed without the Class-Balanced Grouping and Sampling (CBGS) (Zhu et al., 2019) strategy.

Architecturally, the 3D backbone comprises four PointLAM Blocks connected sequentially. Within each PointLAM Block's Bi-Directional Mamba layer, the Mamba blocks are configured with $d\_state = 16$, $d\_conv = 4$, and $expand = 2$. Compression of 3D features to the Bird's-Eye View (BEV) plane is achieved by first dividing the BEV space into regions and then applying random sampling within each region. The subsequent BEV backbone and detection head architectures are consistent with those employed in LION (Liu et al., 2024b) and DSVT (Wang et al., 2023).

## B.2 DOUBLY SORTED SAMPLING

Our Doubly Sorted Sampling (DSS) implementation efficiently selects informative points from the input point cloud. Initially, the cloud is partitioned into uniform regions. For each point, concatenated geometric features ($F_{\text{point}}$) are computed and then processed through the Deviation Network (DevNet) to derive an importance score $S(p_i)$. All points are subsequently subjected to a two-stage sorting process: first globally by their importance score $S(p_i)$ in descending order, and then stably by their assigned region indices. This critical sorting groups points by local region while ensuring that within each region, points are ordered by their importance.

The final selection of the top-$k$ points from the prepared, sorted list is performed with an efficient masking operation, as outlined in Algorithm A. This procedure operates on the `sorted_region_indices` (representing $r'_j$, the region index of point $p'_j$ at 0-indexed position $j$). A boolean `mask` for all $N$ points is initialized to `True` (Line 4), which inherently handles the selection for the first $k$ points in the list (where $j < k$). The core of the selection (Lines 5-7) then updates the latter part of the mask (`mask[k:]`). It achieves this by comparing the region indices of points from position $k$ onwards (`sorted_region_indices[k:]`, i.e., $r'_j$ for $j \geq k$) with the region indices of points $k$ positions prior (`sorted_region_indices[:-k]`, i.e., $r'_{j-k}$). A mismatch (`!=`) signifies that point $p'_j$ is among the top $k$ desired points for its current local region, thus retaining its selection status. This approach efficiently identifies and keeps the $k$ most important points per region, and applying this mask (Line 8) yields the final downsampled, information-rich points.

Table A: Performance comparison on the **test** set of nuScenes dataset. 'C.V.', 'Ped.', 'M.C.', and 'T.C.' denote construction vehicle, pedestrian, motorcycle, and traffic cone, respectively.

| Method | Representation | NDS | mAP | Car | Truck | Bus | Trailer | C.V. | Ped. | M.C. | Bike | T.C. | Barrier |
|---|---|---|---|---|---|---|---|---|---|---|---|---|---|
| PointPillars (Lang et al., 2019) | Pillar | 45.3 | 30.5 | 68.4 | 23.0 | 28.2 | 23.4 | 4.1 | 59.7 | 27.4 | 1.1 | 30.8 | 38.9 |
| PillarNet (Shi et al., 2022) | | 71.4 | 66.0 | 87.6 | 57.5 | 63.6 | 63.1 | 27.9 | 87.3 | 70.1 | 42.3 | 83.3 | 77.2 |
| CenterPoint (Yin et al., 2021) | | 65.5 | 58.0 | 84.6 | 51.0 | 60.2 | 53.2 | 17.5 | 83.4 | 53.7 | 28.7 | 76.7 | 70.9 |
| VoxelNeXt (Chen et al., 2023b) | | 70.0 | 64.5 | 84.6 | 53.0 | 64.7 | 55.8 | 28.7 | 85.8 | 73.2 | 45.7 | 79.0 | 74.6 |
| TransFusion-L (Bai et al., 2022) | | 70.2 | 65.5 | 86.2 | 56.7 | 66.3 | 58.8 | 28.2 | 86.1 | 68.3 | 44.2 | 82.0 | 78.2 |
| FSDv2 (Fan et al., 2024) | | 71.7 | 66.2 | 83.7 | 51.6 | 66.4 | 59.1 | 32.5 | 87.1 | 71.4 | 51.7 | 80.3 | 78.7 |
| LargeKernel3D (Chen et al., 2023a) | Voxel | 70.6 | 65.4 | 85.5 | 53.8 | 64.4 | 59.5 | 29.7 | 85.9 | 72.7 | 46.8 | 79.9 | 75.5 |
| LinK (Lu et al., 2023) | | 71.0 | 66.3 | 86.1 | 55.7 | 65.7 | 62.1 | 30.9 | 85.8 | 73.5 | 47.5 | 80.4 | 75.5 |
| HEDNet (Zhang et al., 2023) | | 72.0 | 67.7 | 87.1 | 56.5 | 70.4 | 63.5 | 33.6 | 87.9 | 70.4 | 44.8 | 85.1 | 78.1 |
| DSVT (Wang et al., 2023) | | 72.7 | 68.4 | 86.8 | 58.4 | 67.3 | 63.1 | **37.1** | 88.0 | 73.0 | 47.2 | 84.9 | 78.4 |
| LION (Liu et al., 2024b) | | 73.9 | 69.8 | 87.2 | 61.1 | 68.9 | 65.0 | 36.3 | 90.0 | 74.0 | 49.2 | 87.3 | 79.5 |
| Voxel Mamba (Zhang et al., 2024b) | | 73.0 | 69.0 | 86.8 | 57.1 | 68.0 | 63.2 | 35.4 | 89.5 | 74.7 | 50.8 | 86.9 | 77.3 |
| 3DSSD (Yang et al., 2020) | Point | 56.4 | 42.6 | 81.2 | 47.2 | 61.4 | 30.5 | 12.6 | 70.2 | 36.0 | 8.6 | 31.1 | 47.9 |
| **PointLAM** (ours) | | 73.0 | 68.8 | 87.8 | 60.8 | 72.4 | 63.2 | 25.7 | 88.6 | 71.4 | 48.8 | 86.3 | 79.3 |

Table B: Performance comparison on the **test** set of Waymo Open Dataset (single-frame setting). Symbol '-' means that the result is not available. "3f" stands for 3-frame model.

| Method | Representation | ALL (3D mAPH) L1 | L2 | Vehicle (AP / APH) L1 | L2 | Pedestrian (AP / APH) L1 | L2 | Cyclist (AP / APH) L1 | L2 |
|---|---|---|---|---|---|---|---|---|---|
| PointPillar (Lang et al., 2019) | Pillar | - | - | 68.6 / 68.1 | 60.5 / 60.1 | 68.0 / 55.5 | 61.4 / 50.1 | - | - |
| PillarNeXt-3f (Li et al., 2023b) | | 79.0 | 74.1 | 83.3 / 82.8 | 76.2 / 75.8 | 84.4 / 81.4 | 78.8 / 76.0 | 73.8 / 72.7 | 71.6 / 70.6 |
| PV-RCNN (Shi et al., 2020a) | Point-Voxel | 74.2 | 68.8 | 80.6 / 80.1 | 72.8 / 72.4 | 78.2 / 72.0 | 71.8 / 66.0 | 71.8 / 70.4 | 69.1 / 67.8 |
| PV-RCNN++ (Shi et al., 2023) | | 75.7 | 70.2 | 81.6 / 81.2 | 73.9 / 73.5 | 80.4 / 75.0 | 74.1 / 69.0 | 71.9 / 70.8 | 69.3 / 68.2 |
| CenterPoint (Yin et al., 2021) | | 77.2 | 71.9 | 81.1 / 80.6 | 73.4 / 73.0 | 80.5 / 77.3 | 74.6 / 71.5 | 74.6 / 73.7 | 72.2 / 71.3 |
| AFDetV2 (Hu et al., 2022) | | 75.2 | 70.3 | 80.5 / 80.0 | 73.0 / 72.6 | 79.8 / 74.3 | 73.7 / 68.6 | 72.4 / 71.2 | 69.8 / 69.7 |
| SST-3f (Fan et al., 2022a) | | 78.3 | 72.8 | 81.0 / 80.6 | 73.1 / 72.7 | 83.3 / 79.7 | 76.9 / 73.5 | 75.7 / 74.6 | 73.2 / 72.2 |
| FSDv1 (Fan et al., 2022b) | Voxel | 78.2 | 72.4 | 82.7 / 82.3 | 74.4 / 74.1 | 82.9 / 77.9 | 75.9 / 71.3 | 75.6 / 74.4 | 72.9 / 71.8 |
| FSDv2 (Fan et al., 2024) | | 79.0 | 73.3 | 82.4 / 82.0 | 74.4 / 74.0 | 83.8 / 78.9 | 77.4 / 72.8 | 77.1 / 76.0 | 74.3 / 73.2 |
| SAFDNet (Zhang et al., 2024a) | | 79.8 | 74.6 | 83.9 / 83.5 | 76.6 / 76.2 | 84.3 / 79.8 | 78.4 / 74.1 | 77.5 / 76.3 | 74.6 / 73.4 |
| Voxel Mamba (Zhang et al., 2024b) | | 79.6 | 74.3 | 84.4 / 84.0 | 77.0 / 76.6 | 84.8 / 80.6 | 79.0 / 74.9 | 75.4 / 74.3 | 72.6 / 71.5 |
| **PointLAM** (ours) | Point | 79.8 | 74.4 | 83.2 / 82.9 | 75.5 / 75.2 | 85.2 / 80.7 | 79.4 / 75.2 | 76.9 / 75.7 | 74.1 / 72.9 |

# C  MAIN RESULTS ON TEST SET

To further validate the generalization capability and robustness of PointLAM, we further evaluate our model on the test split and compare it against state-of-the-art methods.

**NuScenes Test Set.** As detailed in Table A, PointLAM demonstrates strong performance, achieving 73.0 NDS and 68.8 mAP. This establishes a new state-of-the-art for point-based methods, surpassing the previous point-based baseline 3DSSD by a substantial margin of +16.6 NDS. Furthermore, PointLAM remains highly competitive against top-tier voxel-based detectors, matching the performance of Voxel Mamba (73.0 NDS) and outperforming DSVT (72.7 NDS).

**Waymo Test Set.** As presented in Table B, PointLAM demonstrates robust generalization on the official Waymo test benchmark, achieving 79.8 L1 mAPH and 74.4 L2 mAPH. This performance is highly competitive with state-of-the-art voxel-based methods, matching SAFDNet (74.6 L2) and slightly surpassing Voxel Mamba (74.3 L2). Notably, our model exhibits superior capability in detecting smaller objects, outperforming Voxel Mamba by +0.3 APH on Pedestrians (L2) and +1.4 APH on Cyclists (L2). This aligns with our design motivation that direct point processing preserves fine-grained geometric details more effectively than voxel quantization.

# D  FURTHER ANALYSIS EXPERIMENTS

This section provides further analysis and ablation studies to rigorously validate the design choices and efficiency of PointLAM. We conduct an in-depth analysis of our Dynamic Point Sampler (DPS), demonstrating the superiority of our importance-based sampling over various baselines and confirming its robustness to key hyperparameters like region size and the number of sampled points ($k$). We also provide additional clarifications on experimental settings, including the fairness of our training schedule and an analysis of Farthest Point Sampling (FPS) implementations, to further substantiate the claims made in the main paper. Collectively, these experiments provide a comprehensive validation of PointLAM's architecture and its superior performance-efficiency trade-off.

Table C: DSS comparison with common point-based downsampling methods.

| Method | Latency (ms) | L1 mAPH | L2 mAPH |
|--------|--------------|---------|---------|
| Random | 3.5 | 60.64 | 57.78 |
| Uniform | 4.1 | 61.76 | 58.96 |
| FPS | 4489.1 | - | - |
| DSS | 4.5 | **75.14** | **69.02** |

Table D: Latency comparison of PyTorch-based vs. CUDA-based (`torch_cluster`) FPS implementations with varying input sizes.

| Input Points | Sampled Points | Latency (ms) | |
|--------------|----------------|--------------|------------------|
| | | torch_cluster | PyTorch-based |
| 10,000 | 2,000 | **40** | 287 |
| 50,000 | 10,000 | **948** | 1271 |
| 100,000 | 20,000 | 3755 | **2557** |
| 200,000 | 40,000 | 14951 | **4472** |
| Waymo | Waymo | 14510 | **4489** |

## D.1 FURTHER ANALYSIS ON THE DYNAMIC POINT SAMPLER (DPS)

**Comparison with Point-based Downsampling Methods.** To validate the performance of our Doubly Sorted Sampling (DSS) method, we benchmark it against three widely-recognized point downsampling techniques, with results summarized in Table C. As shown, simple methods like Random and Uniform Sampling are fast (3.5ms and 4.1ms, respectively), but their failure to preserve crucial geometric information leads to poor detection accuracy.

Conversely, Farthest Point Sampling (FPS), the de facto standard for ensuring uniform coverage in many point-based models, proves to be computationally prohibitive for large-scale scenes. Its iterative, serial nature results in a latency of over 4400ms—more than three orders of magnitude slower than other methods—making it entirely impractical for training and real-time inference on datasets like Waymo.

In contrast, our DSS method achieves the best of both worlds. It delivers a massive leap in accuracy to 75.14 L1 mAPH, a gain of about 15 points compared to the fast baselines. More importantly, it achieves this with a latency of only 4.5ms, which is comparable to the fastest methods and approximately 1000x faster than FPS. This result demonstrates that DSS effectively breaks the long-standing trade-off between sampling speed and representativeness, enabling both high-fidelity point selection and real-time performance.

**Note on FPS Implementations.** To provide a comprehensive analysis of the Farthest Point Sampling (FPS) bottleneck, we benchmarked two common implementations: a standard PyTorch-based version and the highly optimized CUDA-based version from the `torch_cluster` library. The results, summarized in Table D, reveal a critical scaling behavior.

While the CUDA-based implementation is significantly faster for smaller point clouds (e.g., < 50,000 points), the PyTorch-based version becomes substantially more efficient as the scene size grows. For a typical Waymo point cloud, our PyTorch-based FPS is over 3.2x faster than the specialized CUDA version (4,489ms vs. 14,510ms). We hypothesize that the CUDA kernel's fixed overhead or memory access patterns scale less favorably than the large-scale, batched tensor operations leveraged by the PyTorch implementation on massive inputs. This analysis demonstrates that regardless of the implementation, FPS remains a prohibitive computational bottleneck for high-resolution point clouds, underscoring the critical need for an efficient alternative.

## D.2 ANALYSIS OF TRAINING EFFICIENCY ON NUSCENES

This section clarifies the training settings used for the nuScenes benchmark to provide a fair comparison of overall training cost. While our model is trained for 36 epochs and several leading competitors (e.g., DSVT, HEDNet, SAFDNet) report 20-epoch schedules, a direct comparison of epoch counts is misleading due to the use of the Class-Balanced Grouping and Sampling (CBGS) (Zhu et al., 2019) strategy in their pipelines.

A quantitative analysis of the total training iterations reveals a crucial distinction. As detailed in their official implementations, the 20-epoch schedules with CBGS enabled result in a total of 154,480 training iterations (7,724 iterations/epoch × 20 epochs). In contrast, our 36-epoch schedule, trained without CBGS, amounts to only 63,324 total iterations (1,759 iterations/epoch × 36 epochs).

This calculation demonstrates that PointLAM achieves its state-of-the-art NDS score using approximately 41% of the total training iterations required by these competing methods. This not only confirms the fairness of our comparison but also highlights a significant and previously unstated advantage of our approach: superior data efficiency and faster convergence.

