# OpenReview forum: "PointLAM: Local Attentive Mamba for Efficient Point-based 3D Object Detection"
_ICLR.cc/2026/Conference — Submitted to ICLR 2026_

### Official Review · Reviewer_1dDG · 2025-10-21

**Soundness:** 3
**Presentation:** 3
**Contribution:** 3
**Rating:** 4
**Confidence:** 3

**Summary:**

The paper presents PointLAM for point-based 3D object detection. First,  the paper proposes Dynamic Point Sampler (DPS) that curates an information-rich and structurally representative subset of raw points. Second,  synergizes Bi-Directional Mamba (BDM) layers for global context modeling, and  Local Multiplicative Aggregation (LMA) layers for efficiently capturing intricate local geometries.

**Strengths:**

1. The paper presents a couple of modules including Dynamic Point Sampler (DPS), Doubly Sorted Sampling (DSS), Local Multiplicative Aggregation (LMA) layer Bi-Directional Mamba (BDM) layers to capture intricate geometric patterns, improve efficiency, and enhance the overall performance.
2.  PointLAM achieves competitive performance on large-scale 3D object detection benchmarks including nuScenes and Waymo. As a point-based method, it is even more efficient than the strong and highly efficient voxel-based competitors LION and DSVT, with only a fraction of their complexity in terms of parameters, operations, and latency costs.

**Weaknesses:**

1. The key contribution seems to be efficient implementation details, which are not very fundamental. These contributions seem somehow empirical and the paper lacks deep analysis and key concept.
2. The overall performance is just comparable with or fall behind existing works. Only efficiency is improved.
3. The authors do not provide qualitative comparisons. I think such comparsions will help readers to better understand the key contirbutions.

**Questions:**

1. Why do you evaluate on Waymo and nuScenes validation set, not test set?

---

> ### Author Response · Authors · 2025-11-27
> **[Part 1] Official Comment by Authors**
>
> We appreciate the reviewer’s insightful comments. Please refer to our revised paper for the newly added results and corresponding visualizations.
>
> **W1:**
>
> PointLAM represents a fundamental architectural shift designed to solve the "efficiency-precision dilemma" in point-based 3D detection.
>
> 1. Theoretical Analysis
>
> Our Deviation Network (DevNet) is a theoretical shift from Low-Pass Filtering to High-Pass Filtering. Standard VFE performs max-pooling, which suppresses local variations (high-frequency signals like edges) to retain a dominant feature. This is structurally blind to fine-grained geometry. DevNet approximates a Discrete Laplacian Operator, explicitly amplifying feature deviations ($h_i - \bar{h}$) relative to the local mean. This ensures a Dense Gradient Flow where every point contributes to optimization, fundamentally resolving the information loss inherent in voxel quantization. For a rigorous theoretical analysis, please kindly refer to our Response to Reviewer T4SB on W1.
>
> 2. Algorithmic Transformation
>
> We decrease the computational complexity of sampling. Traditional point-based methods rely on Iterative algorithms (FPS), which are fundamentally serial and scale quadratically ($O(N^2)$) or log-linearly with high constants. DPS introduces a Parallel sorting-based paradigm (DSS), reducing complexity to $O(N \log N)$. This is a fundamental algorithmic contribution that makes point-based sampling feasible for large-scale scenes ($>100k$ points).
>
> 3. Architectural Decoupling
>
> By designing the LMA module to handle local topology via Transient Grid Indexing (an $O(N)$ operation avoiding explicit Graph Search), we relieve the backbone from needing complex serialization (like Hilbert curves) to learn local geometry. This allows the use of simple, highly efficient Axis Sorting for global context, proving that complex inductive biases are unnecessary when the architecture is correctly decoupled.
>
> **W2:**
>
> In the safety-critical domain of autonomous driving, achieving competitive accuracy within strict latency and resource constraints is a primary objective.
>
> 1. Closing the Gap with Voxel Baselines:
>
> While our performance is comparable to leading voxel-based methods, this represents a significant milestone for the point-based paradigm. Historically, point-based methods have lagged behind voxel counterparts due to efficiency bottlenecks. As shown in the new nuScenes Test Set results (Table A) , PointLAM achieves 73.0 NDS, effectively matching Voxel Mamba (73.0 NDS) and outperforming DSVT (72.7 NDS). This demonstrates that a pure point-based model can rival heavy voxel-based systems, closing the long-standing performance gap.
>
> 2. Critical Importance of Efficiency:
>
> Efficiency is not merely an optimization but a fundamental requirement for real-world deployment. On-board hardware in autonomous vehicles operates under strict latency budgets (to ensure timely reaction) and limited energy/computational resources. PointLAM achieves this competitive accuracy while using approximately 50% fewer parameters and FLOPs compared to competitors like LION and running 2x faster. This substantial reduction in computational overhead frees up critical resources for other perception and planning tasks, making PointLAM a highly practical solution for production systems.

---

> ### Author Response · Authors · 2025-11-27
> **[Part 2] Official Comment by Authors**
>
> **W3:**
>
> To provide qualitative insight into our key contributions, we have added **Figure 4** in the revised manuscript.
>
> 1. Visualization of DPS Selectivity: The figure contrasts raw input points (light gray) with the subset retained by our Dynamic Point Sampler (colored). It clearly demonstrates that DPS effectively acts as a geometric filter, aggressively discarding non-informative points on flat road surfaces while preserving the complete structural skeleton of foreground objects.
>
> 2. Visualization of LMA Feature Learning: The retained points are colored based on the magnitude of features extracted by the LMA module (Red: High, Blue: Low). The heatmaps reveal a distinct geometry-aware pattern: high activations are densely concentrated on structural boundaries, edges, and corners, whereas feature responses on flat background surfaces are effectively suppressed. This qualitatively validates that PointLAM prioritizes geometrically salient regions for efficient detection.
>
> **Q1:**
>
> We utilized validation sets as they are the standard benchmarks for ablation studies and method development in this field. To address your concern regarding generalization, we have now included the official Test Set evaluation in the revised manuscript.
>
> - nuScenes Test Set: As shown in **Appendix Table A**, PointLAM achieves 73.0 NDS and 68.8 mAP, confirming its strong performance on the test split. This significantly outperforms the previous point-based baseline 3DSSD by +16.6 NDS and demonstrates performance comparable to top-tier voxel-based detectors like Voxel Mamba (73.0 NDS) and DSVT (72.7 NDS).
> - Waymo Test Set: As shown in **Appendix Table B**, PointLAM demonstrates robust scalability, achieving 74.4 L2 mAPH. It effectively ties with state-of-the-art voxel detectors like SAFDNet (74.6) and Voxel Mamba (74.3). Notably, PointLAM outperforms Voxel Mamba on Cyclists (+1.4 APH) and Pedestrians (+0.3 APH), validating that our point-based architecture preserves fine-grained geometric details more effectively than voxel quantization.

---

### Official Review · Reviewer_T4SB · 2025-10-25

**Soundness:** 3
**Presentation:** 2
**Contribution:** 2
**Rating:** 4
**Confidence:** 4

**Summary:**

This paper presents PointLAM, a point-based 3D detector that operates directly on raw point clouds without voxelization. The method introduces two modules—DPS for sampling distinctive point features and DSS for preserving informative points through the backbone—and integrates a Mamba-based component to further boost performance. Experiments and ablations demonstrate the effectiveness of these designs.

**Strengths:**

- `Strong empirical results:` The method delivers competitive performance on nuScenes and Waymo, with gains over prior point- and voxel-based baselines.

- `Effective point-based design:` Operating directly on raw point clouds avoids quantization artifacts. The DPS module and DSS module contribute improvements, and the Mamba-based component further boosts performance.

- `Thorough evaluation:` Experiments and ablations attribute gains to each component, making the source of improvements easy to understand and compare across settings.

- `Clear presentation:` The paper is well written and easy to follow, with a modular architecture and clear descriptions that facilitate reproducibility.

**Weaknesses:**

`Motivation and claimed advantage over voxelization:`
The paper argues for operating on raw points, but in practice both point-based and voxel-based pipelines downsample/aggregate and inevitably discard information. Voxel methods ingest raw points, perform quantization (pillars/voxels), and use VFE/PFE to aggregate; point-based methods also subsample aggressively to keep computation tractable. Without a formal analysis, it’s unclear that PointLAM fundamentally mitigates information loss relative to strong voxel baselines.

`Limited technical novelty; clarity of differences to prior work:`
- Deviation network vs VFE/PFE (e.g., SECOND, PointPillars [1]): The proposed feature aggregation appears conceptually similar to voxel encoders that pool local neighborhoods. Please clarify the concrete differences. An ablation replacing the proposed module with a standard VFE/PFE would help quantify novelty and necessity.

- Doubly Sorted Sampling (DSS) vs F-FPS (3DSSD [2]): DSS seems close to F-FPS or hybrid feature–distance sampling. What is the exact distinctiveness criterion, sorting strategy, and complexity (e.g., O(N log N) vs O(Nk))?

- Bi-directional Mamba vs VisionMamba [3] and DSVT [4]: The Bi-directional mamba part looks related. Please articulate the architectural and algorithmic differences.

[1]. Pointpillars: Fast encoders for object detection from point clouds

[2]. 3dssd: Point-based 3d single stage object detector.

[3]. Vision mamba: Efficient visual representation learning with bidirectional state space model

[4]. Dsvt: Dynamic sparse voxel transformer with rotated sets


`Baseline coverage for neighbor search:` The voxel-grid–based neighbor query is interesting, but it would be more compelling with direct comparisons to alternative neighborhood constructions, especially Point Transformer v3 (PTv3) [5]. Please report accuracy and efficiency compared to the neighboring search method listed in [5].

[5]. Point transformer v3: Simpler faster stronger

**Questions:**

N/A

---

> ### Author Response · Authors · 2025-11-27
> **[Part 1] Official Comment by Authors**
>
> We appreciate the reviewer’s insightful comments. Please refer to our revised paper for the newly added results and corresponding visualizations.
>
> **W1:**
>
> We provide a formal analysis demonstrating that PointLAM mitigates the irreversible information loss inherent to voxel baselines through distinct signal operators and optimization dynamics:
>
> 1. Operator Analysis: Discrete Laplacian vs. Low-Pass Filtering
>
>     Standard VFE performs max-pooling aggregation, $\mathbf{f}_{VFE} = \max _{i \in \Omega} (W \mathbf{x}_i)$, which functionally acts as a Low-Pass Filter, suppressing local variations to retain only the dominant signal. In contrast, our DevNet computes the feature deviation $\delta_i = \mathbf{h}_i - \frac{1}{N}\sum \mathbf{h}_j$. From a Graph Signal Processing perspective, this approximates a **Discrete Laplacian Operator** (High-Pass Filter). By explicitly amplifying the high-frequency geometric components (e.g., edges, corners) relative to the local mean, DevNet preserves geometrically distinctive points regardless of their absolute feature magnitude, whereas voxelization suffers from spatial quantization loss.
>
> 2. Gradient Flow: Dense Optimization vs. Feature Masking
>
>     Voxel-based max-pooling induces a sparse, "Winner-Take-All" gradient flow. For any non-maximal point $\mathbf{x}_{non-max}$, the gradient $\frac{\partial \mathcal{L}}{\partial \mathbf{x}} = 0$, causing a Feature Masking effect where weak foreground signals are strictly suppressed by strong background features within the same voxel. Conversely, PointLAM’s mean-subtraction mechanism ensures a **Dense Gradient Flow**. The gradient for any point $p_i$ is derived as:
>
>     $$\frac{\partial \mathcal{L}}{\partial \mathbf{h}_i} = \frac{\partial \mathcal{L}}{\partial \delta_i} \left( 1 - \frac{1}{N} \right) - \frac{1}{N} \sum _{j \ne i} \frac{\partial \mathcal{L}}{\partial \delta_j}$$
>
>     This formulation forces the network to learn cooperative contrast, optimizing weights to maximize the statistical distinctiveness of salient points ($p_i$) relative to their neighborhood ($\delta_j$). This resolves spatial ambiguity by ensuring every point contributes to the optimization, avoiding the channel squeeze inherent in VFE.
>
> 3. Experimental Validation
>
>     The theoretical advantage of this high-pass, dense-gradient approach is substantiated by the ablations in **Table 4**. The baseline using standard Point Feature Network (PFN) and Pooling—representative of voxel-based aggregation—achieves 67.41mAP and 71.42 NDS. Replacing PFN with our Laplacian-based DevNet improves performance to 67.82mAP and 71.59 NDS, confirming the efficacy of deviation encoding. Furthermore, combining DevNet with DSS reaches 68.14mAP and 71.82 NDS, demonstrating that our content-aware sampling significantly outperforms solely aggregation baselines.
>
>
> **W2.1:**
>
> To rigorously quantify the novelty and necessity of our design, we map the structural differences directly to our ablation study in **Table 4**:
>
> - Point Feature Calculation: Standard VFE (represented as 'PFN' in Row 1) employs a PointNet-like structure, concatenating the max-pooled feature of the voxel to each point to encode context. In contrast, our DevNet (Row 2) replaces this with a lightweight deviation operator ($h_i - \mu_{\Omega}$), which explicitly encodes the high-frequency "geometric sharpness" of each point rather than the dominant signal.
> - Output Operation: VFE (Row 1: 'Pooling') aggregates all points into a single quantized voxel vector. Conversely, our DSS (Row 4) replaces aggregation with selection, outputting the top-$k$ points with continuous coordinates based on the calculated deviation scores.

---

> ### Author Response · Authors · 2025-11-27
> **[Part 2] Official Comment by Authors**
>
> **W2.2:**
>
> While both methods aim to select informative points, they fundamentally differ in their dependency mechanism and computational paradigm:
>
> 1. Distinctiveness Criterion:
>     - F-FPS (Dynamic Relative Distance): The criterion is dynamic. It selects point $p_j$ that maximizes the feature distance to the already selected set $\mathcal{S}$: $\max_{j} (\min_{p_i \in \mathcal{S}} D(f_j, f_i))$. This necessitates recalculating distances after every selection.
>     - DSS (Static Absolute Saliency): The criterion is *static*. We calculate an independent importance score $S(p_i) = ||F'_{p_i}||_2$ based on the feature deviation computed by DevNet. Once computed, this score does not change, allowing for parallel selection.
> 2. Strategy: Sorting vs. Iteration
>     - F-FPS (Serial Iteration): Must be executed sequentially for $k$ steps (picking one point at a time).
>     - DSS (Parallel Sorting): Employ a non-iterative "Global Sort + Stable Sort" strategy. We first sort all points by score $S(p_i)$, then stably sort by region index. The top-$k$ points per region are then selected simultaneously via a single masking operation.
> 3. Complexity & Efficiency
>     - F-FPS: $O(N \times k)$ or $O(N^2)$. The iterative dependency makes it computationally prohibitive for large-scale clouds.
>     - DSS: $O(N \log N)$. Dominated solely by the sorting operation.
>     - Experimental Proof: As shown in Appendix **Table C**, this theoretical difference results in a massive latency gap on Waymo: DSS (4.5ms) is $\sim 1000\times$ faster than FPS (4489ms).
>
> **W2.3:**
>
> We clarify that we do not claim the Bi-directional Mamba (BDM) block itself as a standalone architectural invention. Instead, our contribution lies in the adaptation of this linear-complexity block to unordered large-scale 3D point clouds and, crucially, its synergy with our proposed LMA module to form an efficient backbone.
>
> 1. Ours Vs. VisionMamba (Synergy & Data Domain):
>
> - Architecture: VisionMamba relies on a "Pure SSM" architecture. In contrast, PointLAM employs a "Global + Local" hybrid design. We identified that Mamba alone struggles with fine-grained 3D geometry; thus, we pair BDM (global context) with our novel LMA (local geometry) layers. **Table 5** confirms this synergy yields a +2.24 NDS gain compared to using BDM alone.
> - Data: VisionMamba processes dense 2D images. PointLAM adapts Mamba to sparse, unstructured 3D points via a specific axis-sorting serialization strategy.
>
> 2. Ours Vs. DSVT (Complexity & Receptive Field):
>
> - Complexity: DSVT is a Transformer-based method using Window Attention, which entails quadratic complexity $O(N_w^2)$ within local windows. BDM achieves strictly linear complexity $O(N)$.
> - Algorithm: DSVT requires partitioning points into windows, limiting the receptive field. BDM processes the entire scene sequence in a single pass, enabling a global receptive field without complex window-shifting operations.
>
> **W3:**
>
> We have added a direct comparison with the serialization-based neighbor search of Point Transformer v3 (PTv3) in **Table 10**. Our grid-based query significantly outperforms PTv3, achieving higher accuracy (75.63 vs. 72.52 L1 mAPH) with dramatically lower latency (2.7ms vs. 11.2ms). While PTv3 is faster than KNN, its reliance on computing space-filling curves (e.g., Hilbert) and reordering points still incurs noticeable overhead. In contrast, our LMA utilizes transient grid indexing to map neighbors in $O(N)$ time, maximizing efficiency while preserving precise local geometry.

---

### Official Review · Reviewer_2xSG · 2025-10-26

**Soundness:** 2
**Presentation:** 2
**Contribution:** 2
**Rating:** 4
**Confidence:** 3

**Summary:**

This paper introduces PointLAM, a novel point-based 3D object detection framework that addresses the efficiency-precision trade-off in LiDAR-based detection. The method combines two key innovations: (1) Dynamic Point Sampler (DPS) with Deviation Network (DevNet) and Doubly Sorted Sampling (DSS) for intelligent point downsampling, and (2) PointLAM blocks that synergize Bi-Directional Mamba (BDM) for global context and Local Multiplicative Aggregation (LMA) for local geometry modeling. The approach achieves competitive performance with voxel-based methods while maintaining superior efficiency.

**Strengths:**

1) PointLAM achieves competitive performance on both nuScenes (72.2 NDS) and Waymo (73.6 L2 mAPH) datasets, matching or exceeding strong voxel-based competitors like LION and DSVT while using significantly fewer parameters and achieving faster inference.
2) The paper provides thorough comparisons across multiple metrics, datasets, and efficiency measures (parameters, FLOPs, latency), demonstrating the method's practical advantages.

**Weaknesses:**

1) While the combination is novel, individual components are relatively incremental. The Deviation Network is essentially a simple feature difference operation, and BDM uses standard axis-based serialization without significant innovation over existing Mamba adaptations for 3D data.
2) The paper lacks comprehensive ablation studies on key components. What happens with different deviation formulations? How sensitive is performance to the choice of k in DSS? The impact of different serialization strategies in BDM is mentioned but not thoroughly analyzed.

**Questions:**

1) How does the method perform when the point cloud density varies significantly across the scene?
2) What is the computational overhead of the temporary voxelization in LMA compared to direct neighborhood queries?
3) How sensitive is the method to the hyperparameter choices (k in DSS, kernel size in LMA)?

---

> ### Author Response · Authors · 2025-11-27
>
> We appreciate the reviewer’s insightful comments. Please refer to our revised paper for the newly added results and corresponding visualizations.
>
> **W1:**
>
> We respectfully argue that the "simplicity" of our components is a deliberate design choice tailored for the strict latency constraints of autonomous driving.
>
> 1. Design Philosophy: Efficiency by Construction
>
> In large-scale detection (processing 100k+ points), "Simple & Effective" is the highest priority. Our contribution lies in identifying the minimal efficient operators needed to achieve SOTA performance, proving that complex, heavy modules are not prerequisites for high-performance 3D detection.
>
> 2. Deviation Network (Theoretical Analysis)
>
> While mathematically simple ($h_i - \bar{h}$), DevNet fundamentally differs from standard encoders. From a Signal Processing perspective, it acts as a Discrete Laplacian Operator (High-Pass Filter), explicitly amplifying high-frequency geometric signals (edges, corners). Optimally, it induces a Dense Gradient Flow where every point contributes to feature learning, avoiding the information loss caused by the sparse "Winner-Take-All" gradient in standard VFE (Max-Pooling). For a rigorous theoretical analysis, please kindly refer to our Response to Reviewer T4SB on W1.
>
> 3. Bi-Directional Mamba (Architectural Decoupling)
>
> We employ simple Axis-based serialization due to Architectural Decoupling. Since our LMA module efficiently handles local geometry, the backbone is relieved of the burden of preserving local topology via complex serialization. This allows us to use simple Axis Sort to achieve comparable performance to complex Hilbert curves but at 50x faster speed (**Table 8**). For a detailed architectural comparison with VisionMamba and DSVT, please kindly refer to our Response to Reviewer T4SB on W2.3.
>
> **W2 and Q3:**
>
> We clarify that ablations for the choice of $k$ in DSS and serialization strategies in BDM were included in the initial submission, and we have added new experiments for deviation formulations and kernel sizes to complete the analysis.
>
> 1. Different Deviation Formulations
>
> We investigated different operators for computing point distinctiveness in **Table 9**. The Subtraction mode ($h_i - \bar{h}$) yields the highest performance (75.63 L1 mAPH), outperforming Addition (75.54) and Multiplication (75.49). This confirms that explicitly modeling the feature offset is crucial for capturing the geometric saliency required for informative sampling, whereas other operators tend to blur the distinction between the point and its context.
>
> 2. Sensitivity to $k$ in DSS
>
> As analyzed in **Table 7**, we evaluated the model's sensitivity to the number of sampled points per region ($k \in \{1.0, 1.5, 2.0, 2.5\}$). The performance remains highly stable (L2 mAPH varies only between 68.79 and 69.02). This demonstrates that our DSS strategy is robust to hyperparameter choices, functioning effectively as a plug-and-play module.
>
> 3. Impact of Serialization Strategies
>
> We provided a thorough comparison of serialization strategies (Random, Hilbert, Z-order, Axis Sort) in **Table 8**. While complex curves like Hilbert preserve locality slightly better, they are 50x slower (5.5ms vs. 0.1ms). Simple Axis Sort achieves comparable accuracy because our LMA module effectively captures local geometry beforehand, decoupling the backbone from the need for complex serialization to learn local topology.
>
> 4. Kernel Size in LMA
>
> We analyzed kernel sizes in **Table 11**. Increasing the kernel from $3\times3$ to $7\times7$ brings only marginal gains ($\sim$0.4 L1 mAPH) but increases latency. We adopt $3\times3$ as the default to maintain optimal efficiency.
>
> **Q1:**
>
> We performed a robustness test by simulating varying point densities via uniform interval sampling. As shown in **Table 12**, PointLAM consistently outperforms the state-of-the-art voxel-based detector LION across all density levels (from 1/2 to 1/16). This confirms that our DPS mechanism preserves critical geometric information more effectively than voxelization, especially in sparse scenarios where blind quantization leads to significant information loss.
>
> **Q2:**
>
> We compared the computational overhead in **Table 10.** Our transient grid indexing takes only 2.7ms, whereas explicit K-NN queries take 44.5ms and the serialization-based search (PTv3) takes 11.2ms. The overhead of temporary voxelization is negligible compared to the quadratic complexity of neighbor searching, validating the efficiency of our LMA design.

---

### Official Review · Reviewer_zq3Q · 2025-11-03

**Soundness:** 2
**Presentation:** 3
**Contribution:** 3
**Rating:** 6
**Confidence:** 3

**Summary:**

This paper introduces PointLAM, a novel point-based 3D object detection framework that effectively balances computational efficiency with geometric precision in 3D object detection. Its main contributions include:
1.	A Dynamic Point Sampler (DPS) that intelligently selects informative points via a Deviation Network (DevNet) and Doubly Sorted Sampling (DSS), reducing computational load while preserving structural details.
2.	A Local Multiplicative Aggregation (LMA) module that efficiently captures fine-grained local geometries without expensive neighborhood queries, combined with Bi-Directional Mamba (BDM) layers for global context modeling with linear complexity.
The experiment in this paper is thorough, benchmarking PointLAM on nuScenes and Waymo datasets against state-of-the-art voxel-based (e.g., LION, DSVT) and point-based methods. It demonstrates superior or competitive performance in accuracy (NDS/mAP) while significantly reducing parameters, FLOPs, and latency.

**Strengths:**

The paper presents PointLAM, an innovative point-based 3D object detection framework that achieves an impressive balance between computational efficiency and geometric fidelity. Its originality lies in revisiting the point-based paradigm—long overshadowed by voxel-based methods—and successfully overcoming its inefficiency through two synergistic designs: the Dynamic Point Sampler (DPS) and the Local Multiplicative Aggregation (LMA) layer. The DPS, with its Deviation Network and Doubly Sorted Sampling strategy, introduces a novel feature-based approach for point selection, effectively addressing the classic bottleneck of expensive or lossy sampling. The LMA module further contributes by providing an elegant, lightweight means to model local geometries without explicit neighborhood queries, complemented by Bi-Directional Mamba layers for efficient global context modeling.
The technical quality of the paper is moderate, with clear algorithmic formulations, thorough ablations, and strong empirical validation across nuScenes and Waymo benchmarks.
In terms of significance, PointLAM redefines the feasibility of efficient point-based detection and could inspire a new research direction focusing on lightweight, direct point processing architectures.

**Weaknesses:**

This paper presents a strong contribution, but several weaknesses could be addressed to further solidify its impact.
1.	While the efficiency gains are impressive, the analysis of the trade-offs introduced by the novel LMA module remains somewhat superficial. The LMA's reliance on a transient voxel grid is a clever trick to avoid k-NN, but it inherently reintroduces quantization, which the paper initially criticizes in voxel-based methods. The paper would be strengthened by a deeper investigation into this apparent contradiction. For example, an analysis of how the performance on very small or thin objects (which are most susceptible to quantization artifacts) compares to a baseline with explicit k-NN would be highly informative.
2.	The presentation of this paper needs to be improved. For example, the algorithmic submodules shown in Figure 3 cannot be easily matched to the overall framework based on the textual description, and the caption of the figure could be improved for better clarity.

**Questions:**

1.	Novelty over Prior Mamba-Based Methods:
PointLAM claims to be the first efficient point-based Mamba detector, but prior works like PointMamba and Mamba3D also address point-level modeling. Could the authors elaborate more concretely on what differentiates PointLAM from these in terms of both architecture and theoretical motivation?
2.	Visualization and Interpretability:
Could the authors provide visualizations of the sampled points or learned local feature maps to better interpret what DPS and LMA actually focus on?

---

> ### Author Response · Authors · 2025-11-27
> **[Part 1] Official Comment by Authors**
>
> We appreciate the reviewer’s insightful comments. Please refer to our revised paper for the newly added results and corresponding visualizations.
>
> **W1:**
>
> We clarify that our design fundamentally avoids the quantization artifacts found in standard voxel methods, as supported by our conceptual distinction and specific experiments on small objects.
>
> 1. Conceptual Distinction: Indexing vs. Pooling.
>
> The "quantization artifacts" in voxel-based methods stem from feature pooling (e.g., max-pooling), which merges distinct points into a single voxel feature, causing information loss. In contrast, LMA uses the transient grid solely as a structural scaffold for efficient indexing.
>
> 2. Empirical Verification on Small Objects.
>
> We compared LMA against an explicit K-NN baseline across varying object sizes in **Table 13**. LMA outperforms K-NN, with the largest gain observed on the smallest objects (Size 1.0: 3.75 vs. 2.51 L1 mAPH). This proves LMA suffers little quantization loss. Conversely, it highlights a flaw in K-NN: enforcing a fixed neighbor count ($K$) on sparse, small objects forces the inclusion of distant background noise. LMA’s fixed spatial range grid query avoids this, preserving pure feature for fine-grained targets.
>
> **W2:**
>
> We have extensively revised the figure and the corresponding descriptions.
>
> - Unified Terminology: We renamed "X-Mamba/Y-Mamba" to "BDM-X/BDM-Y" in Figure 3 to strictly align with the "Bi-Directional Mamba (BDM)" terminology used in the text.
> - Clarified Visual Guides: We modified the visual indicators, using solid arrows exclusively for data flow and dashed lines for explanatory references, making the pipeline easier to trace.
> - Improved Correspondence: We rewrote both the figure caption and Section 3.1 ("Overall Architecture"). The descriptions now strictly follow the logical data flow and explicitly map the textual descriptions to the specific visual panels (e.g., Top, Bottom-Left), ensuring matched figures and text.

---

> ### Author Response · Authors · 2025-11-27
> **[Part 2] Official Comment by Authors**
>
> **Q1:**
>
> They diverge fundamentally in Theoretical Motivation (Task Scope) and Architectural Paradigm (Data Topology and Sampling).
>
> 1. Theoretical Motivation:
>
> - Prior Works (Small-Scale Classification): *PointMamba* [1] and *Mamba3D* [2] target indoor objects (\~1k-2k points). At this scale, computationally heavy operations like Farthest Point Sampling (FPS) and k-NN graph construction are tolerable (\~10ms).
> - PointLAM (Large-Scale Detection): We target outdoor scenes (~100k+ points). Empirical analysis in **Table C** dictates that $O(N^2)$ or iterative algorithms (like FPS) create prohibitive latency (>1000ms) at this scale. Our motivation is to prove that linear-complexity, parallelizable architectures can achieve SOTA detection without relying on these heavy legacy operators.
>
> 2. Architectural Differentiation:
>
> - A. Sampling Architecture: Iterative vs. Parallel (DPS)
>     - *PointMamba / Mamba3D:* Rely on FPS, an iterative, serial algorithm where point selection depends on the previous step. This is an architectural bottleneck.
>     - *PointLAM:* Introduces DPS, a fully parallelizable, learnable sampling architecture. By restructuring sampling as a "Score-then-Sort" parallel operation, we change the time complexity class, enabling real-time processing of massive clouds.
> - B. Topology Modeling: Graph Search vs. Transient Grid (LMA)
>     - *Mamba3D:* Relies on k-NN to construct local graphs for its "LNP" module [2]. This requires explicit neighbor search, which is computationally expensive for large-scale data.
>     - *PointLAM:* Replaces graph searching with a Transient Voxel Grid (LMA). We fundamentally change the data interaction mechanism from "Searching" (finding neighbors via distance) to "Indexing" (finding neighbors via hash/voxel mapping). This allows for local aggregation with $O(N)$ complexity, avoiding the irregularity of k-NN.
> - C. Serialization Strategy: Implicit vs. decoupled (BDM)
>     - *PointMamba:* Heavily relies on complex Hilbert Curves [1] to embed 3D locality into 1D sequences because it lacks explicit efficient local modules.
>     - *PointLAM:* Adopts a decoupled architecture. Since our LMA module explicitly handles local geometry, our global learner (BDM) is relieved of the burden of learning local topology. This allows us to use simple Axis Sort, which is significantly faster than Hilbert generation, proving that complex serialization is unnecessary given a strong local aggregator.
>
> **Q2:**
>
> We have provided **Figure 4** in the revised manuscript to interpret the focus of our modules.
>
> 1. DPS Selectivity: The visualization contrasts raw input points (light gray) with those retained by DPS (colored). It demonstrates that DPS aggressively filters out redundant, flat surfaces (e.g., roads) while selectively preserving the complete geometric skeleton of foreground objects.
>
> 2. LMA Interpretability: The points are colored based on the magnitude of features learned by LMA (Red: High, Blue: Low). The heatmaps reveal a distinct geometry-aware pattern: high activations are concentrated on structural boundaries and corners (e.g., vehicle chassis), whereas responses on flat background surfaces are effectively suppressed.
>
> [1] Liang et al., PointMamba: A Simple State Space Model for Point Cloud Analysis.
>
> [2] Han et al., Mamba3D: Enhancing Local Features for 3D Point Cloud Analysis via State Space Model.

---

### Author Response · Authors · 2025-11-27
**General response**

Dear Reviewers,

We sincerely appreciate your constructive comments and insightful suggestions. In this revised paper, we have conducted extensive new experiments and incorporated the following modifications to address your concerns:

- **Enhanced Presentation (Sec. 3.1 & Fig. 3):** We have extensively revised Figure 3 and the corresponding textual descriptions. The updated figure now strictly aligns with the logical data flow, explicitly mapping visual components to specific modules (DPS, LMA, BDM) to ensure clarity.
- **New Experimental Results:** We have added a series of new experiments to validate our model's performance:
    - **Detailed Module Ablations (Tables 9-11):** We include comprehensive ablations on Deviation formulations, Neighbor Query mechanisms, and Kernel sizes to empirically justify our design choices.
    - **Robustness to Point Density (Table 12):** We demonstrate that PointLAM consistently outperforms the voxel-based SOTA (LION) across varying input sparsity levels (from 1/2 to 1/16 density), verifying the efficacy of our DPS sampling.
    - **Performance on Object Sizes (Table 13):** We provide a breakdown of performance across different object sizes, showing that our LMA module outperforms KNN on small objects, confirming its ability to preserve fine-grained features.
    - **nuScenes Test Set (Table A):** We report performance results on the test split, where PointLAM achieves 73.0 NDS. This significantly outperforms the previous point-based baseline by +16.6 NDS and demonstrates performance comparable to top-tier voxel-based detectors.
    - **Waymo Test Set (Table B):** We report performance results on the test split. PointLAM achieves 74.4 L2 mAPH, demonstrating competitive performance against strong voxel baselines while showing distinct advantages in detecting small objects.
- **Visualization and Interpretability (Fig. 4):** We incorporated qualitative visualizations of DPS sampling results and LMA feature activation maps. It intuitively demonstrates our model's ability to filter non-informative background points while selectively highlighting geometric structures.

---

### Author Response · Authors · 2025-12-03
**Summary of Revisions and Responses**

Dear Area Chair and Reviewers,

Given the constraints on the discussion phase, we provide this summary to assist the Area Chair in evaluating our rebuttal. We have extensively revised the manuscript with new experiments and clarifications that comprehensively address the collective concerns of all reviewers.

1. Summary of Major Revisions (Detailed in General Response)

- Comprehensive Ablations: Added Tables 9–13 (Deviation formulas, Neighbor Query, Kernel sizes, Density robustness, Object Size analysis).
- Visualization: Added Figure 4 to demonstrate interpretability.
- Official Test Set Results: Added Tables A-B (nuScenes Test (73.0 NDS) and Waymo Test (74.4 L2 mAPH)).
- Presentation: Modified Figure 3 and Section 3.1 for clarity.

2. Resolutions of Key Reviewer Concerns

Theme A: Theoretical Depth & "Incremental Novelty" (Addressing Reviewers T4SB, 2xSG, 1dDG)

- Concern: Components (DevNet, BDM) appear simple or incremental.
- Resolution: We argued that "simplicity" is a deliberate design choice for efficiency in autonomous driving.
    - DevNet: We clarified its role as a Discrete Laplacian Operator (High-Pass Filter), inducing a dense gradient flow that is fundamentally superior to the Low-Pass (Max-Pooling) filtering of standard VFE.
    - Architecture: We shifted the sampling paradigm from iterative $O(N^2)$ (FPS) to parallel $O(N \log N)$ (DPS), and introduced Architectural Decoupling (LMA handles local topology $\rightarrow$ BDM can use simple Axis Sort), proving complex inductive biases are unnecessary for high performance.

Theme B: Comprehensive Ablation Studies (Tables 9–13) (Addressing Reviewer 2xSG, T4SB, zq3Q)

- Concern: Need for more extensive validation on component choices, density, and sensitivity.
- Resolution: We added a suite of experiments to rigorously validate our design:
    - Module Choices (Tables 9–11): We empirically justified our choices for Deviation formulations (Subtraction is optimal), Neighbor Query mechanisms (Grid > PTv3 > KNN in efficiency), and Kernel sizes ($3\times3$ balances speed/accuracy).
    - Robustness to Density (Table 12): PointLAM consistently outperforms the voxel-based SOTA (LION) across varying sparsity levels (from 1/2 down to 1/16 density), verifying the robustness of DPS sampling.
    - Object Size Analysis (Table 13): We demonstrated that LMA significantly outperforms KNN on small objects (Size 1.0), confirming its ability to preserve fine-grained features without background noise contamination.

Theme C: "Quantization Artifacts" in LMA (Addressing Reviewers zq3Q, 2xSG, T4SB)

- Concern: LMA's grid might reintroduce quantization loss like voxel methods.
- Resolution: We clarified the conceptual distinction: LMA uses the grid for indexing, not pooling. Each point retains its individual feature vector. We provided empirical proof (Table 13) showing LMA significantly outperforms KNN on small objects (Size 1.0), confirming it preserves fine-grained geometry better than explicit neighbor searches which suffer from feature contamination.

Theme D: Comparison with Prior Mamba Works (Addressing Reviewers zq3Q)

- Concern: Differentiation from PointMamba/Mamba3D.
- Resolution: We articulated the fundamental difference in task scope (Large-scale Detection vs. Small-scale Classification). PointLAM uniquely solves the sampling bottleneck (via DPS) and topology bottleneck (via Grid-LMA) that make prior classification-focused methods prohibitive for outdoor scenes ($>100k$ points).

Theme E: Missing Qualitative & Test Results (Addressing Reviewers 1dDG)

- Concern: Lack of visualizations and official Test Set evaluation.
- Resolution:
    - Test Results: We added official results for nuScenes (Table A) and Waymo (Table B), proving PointLAM establishes a new point-based SOTA (+16.6 NDS over 3DSSD) and matches top-tier voxel methods.
    - Visualization: We added Figure 4, visualizing DPS selectivity (filtering ground points) and LMA activation (highlighting object skeletons), demonstrating the model's "content-aware" focus.

3. Conclusion

PointLAM represents a significant milestone: it is the first work to demonstrate that a pure point-based architecture can achieve performance parity with heavy voxel-based systems while using ~50% fewer parameters and FLOPs. We hope this summary facilitates a positive assessment of our contribution to efficient 3D perception.

Best regards,

The Authors

---

### Meta-Review · Area_Chair_fEQb · 2025-12-08

**Summary:**

In the original reviews, most reviewers raised concerns about the limited novelty and technical contribution of the work, the insufficiency of the experimental validation, and issues with the clarity and presentation of the manuscript. There was no discussion between the reviewers and the authors during the rebuttal phase.

The AC has carefully read the paper, the reviews, and the rebuttal. While some minor issues were addressed, the core concerns regarding the novelty and significance of the contribution remain unresolved. In addition, the substantial new content added in the revision (figures, text, experiments, etc.) would likely require a more extensive revision cycle and another round of review to be properly assessed.

Overall, the paper in its current form is not yet ready for publication at ICLR 2026.

**Reviewer Concerns:**

As mentioned in the summary.

**Reviewer Scores:**

Reviewer zq3Q: 6 (no response during rebuttal)
Reviewer 2xSG: 4 (no response during rebuttal)
Reviewer T4SB: 4 (no response during rebuttal)
Reviewer 1dDG: 4 (no response during rebuttal)

---

### Decision · Program_Chairs · 2026-01-26

Reject